# CAMLLA-YOLOv8n: Cow Behavior Recognition Based on Improved YOLOv8n

**DOI:** 10.3390/ani14203033

**Published:** 2024-10-19

**Authors:** Qingxiang Jia, Jucheng Yang, Shujie Han, Zihan Du, Jianzheng Liu

**Affiliations:** 1College of Artificial Intelligence, Tianjin University of Science and Technology, Tianjin 300453, China; qingxiangjia@yeah.net (Q.J.); 13483211514@163.com (Z.D.); 2Department of Electronics Engineering, Jeonbuk National University, Jeonju 54907, Republic of Korea; shujiejulie@jbnu.ac.kr; 3Core Institute of Intelligent Robots, Jeonbuk National University, Jeonju 54907, Republic of Korea

**Keywords:** cow behavior recognition, YOLOv8n, CAMLLA-YOLOv8n, attention mechanism, Holstein cows

## Abstract

The daily behaviors of Holstein cows, such as standing, grazing, and lying, as well as abnormal behaviors such as estrus, licking, and fighting, are closely related to their physiological health. Accurately identifying these behaviors is of great significance for monitoring the health of dairy cows. For instance, hoof disease generally causes dairy cows to lie down more, while cows in estrus exhibit mounting behavior. This study employs deep learning technology based on computer vision to detect dairy cow behavior. The experimental results demonstrate that this method effectively meets the need for the accurate and rapid identification of Holstein cow behavior in real agricultural environments, which is crucial for improving the economic benefits of farms.

## 1. Introduction

In livestock farming, studying animal behavior is crucial for understanding how animals perceive and respond to their environment [1]. This allows us to use effective techniques to improve their health and welfare on farms. Daily behaviors of dairy cows such as standing, grazing, and lying down, as well as abnormal behaviors such as estrus, licking, fighting, and aggression, are closely related to their physiological health [2]. For example, normal standing, grazing, and lying down behaviors are usually regarded as manifestations of dairy cows being in a comfortable state; abnormal fighting and licking behaviors may indicate health problems or environmental discomfort; and estrus behavior, as a significant external manifestation, is usually difficult to detect in time due to its low frequency, short duration, unclear performance, and lack of fixed position [3]. If a cow in an estrus is not detected, and a signal is not sent out in time, the farm must wait for the cow’s next estrus to occur, which results in the loss of the cow’s breeding opportunity, an increase in the number of empty days, a longer calving interval, and a decrease in the cow’s milk production, thus affecting the income and benefits of dairy farming [4,5]. Therefore, using intelligent means to monitor and analyze cow behavior in real time in actual application scenarios and to identify cow behavior efficiently and accurately is of great significance for a timely understanding of the health status of cows and improving the economic benefits of farms [6].

At present, cow behavior detection methods can be categorized into three main types: traditional staff observation, contact-based sensor detection, and non-contact image recognition. Traditionally, the monitoring of cow behavior mainly relies on staff observation and recording, which is not only time-consuming and inefficient but also highly subjective. In large-scale dairy farms, it is unrealistic to observe every activity of cows in real time, which makes it difficult for manual observation to achieve continuous and 24 h large-scale monitoring [7,8]. Using contact-based sensors to monitor cow behavior data also has its drawbacks. For instance, these sensors typically require animals to wear them in specific places such as collars or ankles to collect movement and physiological data for behavior identification. However, employing contact-based devices poses challenges, and improper placement may result in inaccurate sensor data. Moreover, these devices are prone to damage, especially in younger animals that are more active, impacting the reliability and continuity of monitoring [9,10]. Therefore, automatic and accurate identification of cow behavior has important research significance and practical application value for early detection of problems and timely intervention measures.

In recent years, with the rapid development of computer vision and deep learning technology, the livestock industry has adopted deep learning to solve individual animal identification, tracking animal movement, body part identification, and species classification. People are increasingly interested in using these models to examine the relationship between livestock behavior and related health problems [11]. Based on video image analysis technology, this method enables non-contact, automated, real-time monitoring of cow behavior. For example, the method of using RGB cameras combined with deep learning models has been proven to accurately classify and monitor individual and group behaviors of dairy cows under various environmental and time conditions [12]. Traditional observation-based or sensor-based behavior detection methods are gradually being replaced by more efficient and automated image-based detection methods [13].

Fuentes et al. [12] proposed a recognition method based on spatiotemporal information, using YOLOv3 for frame-level detection, context-based feature extraction, and combining 3D-CNN and optical flow to capture temporal features. It was tested on 15 different cattle behavior datasets recorded in Korean farms, with an average Precision (mAP) of 85.6%, which can effectively identify individual cattle, groups, and partial actions. However, the system showed some difficulties in identifying smaller partial actions and coping with changes in background and lighting conditions (especially at night). Wang et al. [14] used an improved model E-YOLO based on YOLOv8n, which uses normalized Wasserstein distance loss, a context information enhancement module, and a triple attention module to detect estrus behavior in dairy cows. Its has an average Precision of estrus detection of 93.90%, an Average Precision of mounting (APmounting) of 95.70%, and an F1-score of 93.74%, showing high efficiency and accuracy in a real dairy farm environment. However, one limitation of the E-YOLO model is that it only focuses on detecting estrus and does not address other important cow behavior detection issues. Wang et al. [15] proposed an efficient 3D CNN (E3D) algorithm based on the SandGlass-3D module combined with 3D convolution and Dwise (depthwise separable convolution), which is specifically used to identify the basic movement behavior of dairy cows. The model combines the SandGlass-3D module and the efficient channel attention (ECA) mechanism to effectively process the spatiotemporal information in the video, quickly and accurately identify the behavior of dairy cows in the natural environment, and achieve an accuracy of 98.17%. However, the model still faces challenges in running speed and efficiency on mobile devices with limited hardware resources. Yu et al. [16] proposed a DRN-YOLO method based on DenseResNet for the real-time monitoring of dairy cow feeding behavior. By replacing the CSPDarknet backbone network with a self-designed DRNet backbone network, and based on the YOLOv4 algorithm, the multi-scale and spatial pyramid pooling (SPP) structure were used to enhance the interaction of scale semantic features. Compared with YOLOv4, DRN-YOLO improves accuracy, Recall and mAP by 1.70%, 1.82% and 0.97%, respectively. However, it faces low recognition accuracy and insufficient feature extraction in complex dairy farming environments. Bai et al. [17] proposed an improved model, GC_Res2 YOLOv3, based on YOLOv3 for identifying cow behavior. The Res2 network structure and Global Context Block are combined to enhance the model’s multi-scale perception ability in dense scenes. The improved model has an accuracy of 90.6%, 91.7%, 80.7% and 98.5% in detecting the four behaviors of standing, lying, walking and mating in cows. The overall average accuracy is 90.4%. However, although the model shows good generalization ability in a variety of scenarios, it still has confusion in distinguishing the walking and standing behaviors of cows. Wang et al. [18] proposed a lightweight cow mounting behavior recognition system based on improved YOLOv5s. The system uses the concept of EfficientNetV2 to design a lightweight background network and introduces an attention mechanism, inverted residual structure, and depth-separable convolution. The model’s inference speed is as high as 333.3 fps, the inference time for each image is 4.1 ms, and the model’s mAP value is 87.7%, which is 2.1% higher than the mAP value of YOLOv5s. However, the lightweight network may have a decline in feature-extraction capabilities, which may affect the accuracy of the model.

To solve the above problems, we proposed an improved YOLOv8n cow behavior recognition method CAMLLA-YOLOv8n to improve the effect of cow behavior detection. In summary, our contributions can be summarized as follows:(1)We installed high-definition cameras at a dairy farm in Tianjin, China, and collected video data of Holstein cows’ daily behavior for about 110 days. After removing redundant frames, we collected a total of 2418 images, which were annotated into seven behavior categories based on expert classification, with 23,073 boxes labeled using the CVAT tool.(2)We propose an improved YOLOv8n-based behavior recognition method for Holstein cows, named CAMLLA-YOLOv8n, which integrates the Coordinate Attention mechanism into the C2f module to form the C2f-CA module. This enhances the model’s ability to accurately recognize spatial relationships between different cow positions, focus on key areas, and filter background interference.(3)The Multi-layer Mamba-Like Linear Attention (MLLAttention) mechanism was introduced in the P3, P4, and P5 layers of the Neck of the YOLOv8n model to tackle the challenges posed by significant scale variations in cow behavior recognition.(4)The SPPF-GPE module was formed by improving the SPPF module, combining global average pooling and global maximum pooling to enhance the model’s ability to cope with environmental changes and capture key features of cow behavior.(5)Considering the limitations of traditional IoU loss in cow detection, we introduced Shape–IoU to focus on the shape and scale features of the bounding box, improve the alignment between prediction and Ground Truth Boxes, and enhance detection accuracy.

## 2. Materials and Methods

### 2.1. Dataset

#### 2.1.1. Data Source

The experimental video data were collected at a dairy farm in Tianjin, China. The data collection involved Holstein cows, which were housed in six modern cowsheds, accommodating a total of about 3000 dairy cows. Through field surveys, each dairy cowshed area was measured to be approximately 200 m long and 30 m wide, providing an ideal environment to support large-scale field testing and monitoring experiments. Two Tiandy cameras (model TD-H234S), manufactured by Tiandy Technologies Co., Ltd., based in Tianjin, China, were installed on the dairy farm. They were mounted on the support wall of the crossbeam in the double slope cowshed to facilitate real-time data collection and processing. The camera was fixed at a height of about 4.5 m and tilted downward at an angle of about 30°, allowing it to capture the entire area of dairy cow activities. The camera installation position is shown in Figure 1. The recording period was from 20 April 2024 to 14 August 2024, and the daily behavior video data of dairy cows with a time span of about 110 days were successfully collected.

#### 2.1.2. Dataset Construction

In this study, a total of 60 independent video clips containing cow behaviors were manually selected, each of which contains one or more complete cow behavior instances to capture the behavior patterns of cows in different time periods and environments. During the data collection process, we paid special attention to the natural and unnatural behaviors of cows, such as normal behaviors such as grazing, standing, and lying, as well as abnormal behaviors such as estrus, fighting, and licking. All behavioral data were obtained through field shooting to ensure the authenticity and practical value of the data. The length of each clip ranged from 10 to 25 s, the video resolution was 1280 pixels (Horizontal) × 720 pixels (Vertical), and the frame rate was 30 fps/s. The specific parameters of the specific cow behavior dataset are shown in Table 1. The video format is exported as AVI. The video frame extraction technology is used to sample every 5 frames of each video. In order to avoid excessive similarity between adjacent frames, the structural similarity (SSIM) algorithm [19] is used to compare the differences between consecutive frames in the study and delete redundant images. There is no data overlap between different sample sets. Finally, a total of 2418 images were collected, comprising 1935 for the training dataset and 483 for the validation set, with the proportions used for model training and model validation being 80% and 20%, respectively.

Six behaviors of dairy cows, including grazing, standing, lying, licking, mating, and fighting, and the empty state of the cow bed were selected as the research objects. These behaviors are closely related to the health evaluation of dairy cows. Sample pictures of cow behavior in the dataset are shown in Figure 2. The criteria for determining dairy cow behaviors are set as shown in Table 2. The dataset is annotated with 7 categories, with a total of 23,073 boxes annotated. The video clips were carefully classified by experts and completed by the team using the CVAT data annotation tool according to the specific behavioral characteristics of dairy cows. The category distribution and annotation information of the dataset are shown in Figure 3, where Figure 3a shows the distribution of samples of each category in the dairy cow behavior dataset we constructed, among which lying, standing and grazing behaviors are the most common, while estrus, licking and fighting behaviors are less common. This distribution is consistent with the common behavioral patterns of dairy cows in farm environments and reflects the activity patterns of dairy cows in real farm environments. Figure 3b illustrates the size and quantity of the target boxes, providing a reference for optimizing the anchor box sizes and adjusting the detection thresholds based on target sizes. Figure 3c describes the position of the target center point relative to the entire image, reflecting the spatial distribution characteristics of the position preference and behavior of dairy cows in the image. Figure 3d shows the height–width ratio distribution of the cow target box relative to the entire image, reflecting the multi-scale information of the cow.

#### 2.1.3. Data Augmentation

As shown in Figure 4, considering the problems of different scales of cows, uneven lighting, and excessive occlusion in actual farm environments, we expanded the dataset to enhance the generalization ability of the model. The behavior data of dairy cows obtained in dense farming environments have large differences in multi-scale targets and backgrounds, resulting in low generalization ability of the model. Therefore, data enhancement methods are introduced to enrich the multi-scale behavior data and background information of dairy cows. The Mosaic data enhancement method [20] randomly reads 4 images in the training set and performs random rotation, splicing, scaling, and translation operations, then performs color changes in the three color domains of H, S, and V, and then splices them into one image as training data. The effects of different data enhancement methods are shown in Figure 5. Figure 5a is the original sample image, Figure 5b is the enhanced effect after the Mosaic image, and Figure 5c–i are the enhanced effects of a single image after rotation transformation, irregular shearing, brightness change, noise addition, left–right flipping, up–down flipping, and random occlusion transformation. Compared with the single image enhancement method, the image enhanced by Mosaic contains more rich scenes and multi-scale information, increasing the diversity of the dataset. Image flipping simulates cow behavior samples from different perspectives. Image rotation transformation can simulate cow behavior in different directions. In addition, by adjusting the brightness of the image, we can simulate different lighting conditions from daylight to night. The method of adding noise is used to simulate the image quality under low light and bad weather conditions. Finally, random cropping strengthens the model’s ability to recognize local behavioral characteristics of cows, which is the key to understanding complex behavioral patterns. Using data-enhanced datasets to train models indirectly increases the number of samples, accelerates model convergence, and improves the generalization ability of the model.

### 2.2. Methods

In order to accurately identify the behavior of cows, this study selected YOLOv8n as the basic model for cow behavior recognition. Based on the analysis of the dataset characteristics and the YOLOv8n network structure, the CAMLLA-YOLOv8n model was proposed. The CAMLLA-YOLOv8n cow behavior recognition network architecture is shown in Figure 6. The specific improvements are as follows:(1)Application of CA attention mechanism: To solve the challenges of complex background and increased number of cows caused by the Mosaic fusion of the dataset, the CAMLLA-YOLOv8n model integrates the Coordinate Attention mechanism in the C2f module to form the C2f-CA module. This improvement enables the model to more accurately identify and understand the spatial relationship between different cow positions while improving the model’s sensitivity to key areas and filtering capabilities for background interference.(2)Multi-layer MLLAttention mechanism: To address the challenges of recognizing multi-scale and multi-behavioral features of cows, CAMLLA-YOLOv8n introduces the MLLAttention mechanism in the P3, P4, and P5 layers of the YOLOv8n model Neck to solve the challenges of cow behavior recognition due to large-scale changes and improve the accuracy of behavior recognition.(3)Global-pooling-enhanced SPPF-GPE module: Faced with multi-scale cow behavior data, CAMLLA-YOLOv8n innovatively improves the SPPF module to form the SPPF-GPE module. By combining global average pooling and global maximum pooling processing, the recognition of small targets has been optimized, and the model’s ability to respond to environmental changes and capture key parts of cow behavior has been enhanced.(4)Shape–IoU loss optimization: In view of the limitations of traditional IoU loss in cow detection, Shape–IoU is introduced to focus on the shape and scale characteristics of the Bounding Box, improve the matching degree between the Prediction Box and the Ground Truth Box, and enhance detection accuracy.

#### 2.2.1. Cow Behavior Recognition Model Based on YOLOv8n

Accuracy and real-time performance are the main requirements for identifying cow behavior. YOLOv8 is the most advanced SOTA model in the YOLO series. It balances recognition accuracy and detection speed and is a real-time and efficient single-stage target detection algorithm. YOLOv8 has a total of 5 pre-trained models of different sizes, from small to large, namely n, s, m, l and x. The larger the model, the better the detection effect. However, the larger the model, the larger the number of parameters, the slower the running speed, and the more resources required for training. YOLOv8n is the model with the smallest depth in the YOLOv8 series. Compared with other models in the YOLOv8 series, while ensuring a certain recognition accuracy, the parameters and floating-point operations (FLOPs) are significantly reduced, which is suitable for target detection tasks with few categories and simple features.

YOLOv8n consists of four parts: Input, Backbone, Neck and Head. The YOLOv8n model architecture is shown in Figure 7. Input is the initial layer of the neural network, which is responsible for receiving and processing the input image. Backbone consists of Convolutional Block, C2f and SPPF (Spatial Pyramid PoolingFast), which is responsible for learning complex feature representations from input images. The Convolutional Block consists of the Convolutional Layer, activation function SiLU and normalization layer BatchNorm2d. SPPF is used for feature extraction. Although the C2f module is lightweight, it can obtain rich gradient information. YOLOv8n uses VFL Loss as classification loss, DFL Loss+ CIoU Loss as loss function for bounding box regression, and the Task-Aligned Assigner matching method is used for sample matching. Neck connects Backbone and Head, and its design is crucial to the performance of the detection algorithm. Neck adopts a Feature Pyramid Network (FPN) and Path Aggregation Network (PAN) structure, combines features of different scales, and enhances the network feature fusion capability. Head contains three detection branches of different scales, and obtains the optimal detection box through non-maximum suppression (NMS). Head adopts the Anchor-Free idea to directly predict the center point and size of the target object, no longer relying on the preset Anchor, improving the generalization ability of the model on targets of various scales, shapes and proportions.

#### 2.2.2. Coordinate Attention for Cow Behavior Recognition

In order to accurately identify cow behavior in a complex and dense visual environment, this study proposes the CAMLLA-YOLOv8n model to address the challenges of increasing the number of cows and complicating background information caused by Mosaic fusion technology. CAMLLA-YOLOv8n integrates the Coordinate Attention mechanism into the C2f module of the YOLOv8n network, forming the C2f-CA module. The C2f-CA module weights the feature channels and uses the learned inter-channel dependencies. The model can better understand the relationship between cows in different locations and enhance the network’s sensitivity to cow location information, thereby effectively filtering out extraneous background interference.

The core idea of the CA attention mechanism is to embed position information into channel attention so that the lightweight network can focus on a larger area so that the model can better understand the relationship between different positions. A large amount of computational overhead is avoided. Its structure is shown in Figure 8. In Figure 8, C is the number of channels of the input feature map; H and W are the height and width of the input feature map respectively; r is the reduction rate; and XAvgPool and YAvgPool, respectively, refer to one-dimensional horizontal global pooling and One-dimensional vertical global pooling. The CA attention mechanism encodes channel relationships and long-range dependencies through precise position information and is divided into two steps: coordinate information embedding and Coordinate Attention generation.

In order to alleviate the loss of position information caused by 2D global pooling, the channel attention is decomposed into two parallel 1D feature encoding processes to effectively integrate the spatial coordinate information into the generated attention map. The CA attention mechanism first performs two global average poolings on the input feature map in the width and height directions, respectively, to obtain two feature maps that capture the global features in the width and height directions respectively. The above process is shown in Formula (Equation 1).
(1)$$z_{c}=\frac{1}{H\times W}\sum_{i=1}^{H}\sum_{j=1}^{W}x_{c}\left(i,j\right)$$

Among them, $z_{c}$ represents the global feature; $x_{c}\left(i,j\right)$ is the input of the original feature. In detail, first use the pooling kernel of size (H,1) or (1,W) to encode each channel along the horizontal coordinate and vertical coordinate respectively, aggregate the features along the two spatial directions, and obtain a pair of direction-aware feature maps. Therefore, the output of the c-th channel with height h can be expressed as Formula (Equation 2):(2)$$z_{c}^{h}\left(h\right)=\frac{1}{W}\sum_{0\leq i<W}x_{c}\left(h,i\right)$$

Similarly, the output of the c-th channel with a width of w can be expressed as Formula (Equation 3):(3)$$z_{c}^{W}\left(w\right)=\frac{1}{H}\sum_{0\leq j<H}x_{c}\left(j,w\right)$$

Then, after channel-level splicing operations and convolution smoothing processing, a global receptive field and more accurate information representation are generated, as shown in Formula (Equation 4).
(4)$$f=\delta (F_{1}\left(\left[z^{h},z^{w}\right]\right)$$

Among them, $F_{1}$ is the feature map after batch normalization; f is the feature map obtained by Sigmoid activation function; δ is the nonlinear activation function; and $z^{h}$ and $z^{w}$ are the aggregated feature map. Then, it is transformed into a tensor with the same number of channels as the input data through batch normalization and nonlinear mapping, as shown in Formulas (Equation 5) and (Equation 6).
(5)$$g^{h}=\delta \left(F_{h}\left(f^{h}\right)\right)$$
(6)$$g^{w}=\delta \left(F_{w}\left(f^{w}\right)\right)$$
where $f^{h}$ and $f^{w}$ are the two independent tensors that feature map f is divided into along the spatial dimension; $F_{h}$ and $F_{w}$ transform $f^{h}$ and $f^{w}$ into tensors with the same number of channels as the input, respectively; $g^{h}$ and $g^{w}$ are the attention weights of the input feature map in the height and width directions obtained after the above calculation; and δ is the Sigmoid function. Finally, through multiplication weighted calculation on the original feature map, the final feature map with attention weights in the width and height directions is obtained, and the position information is saved in the generated feature map with attention weights. Finally, the two attention maps are then multiplied by the input feature map to enhance the representation ability of the feature map. The final output of Coordinate Attention is shown in Formula (Equation 7).
(7)$$y_{c}\left(i,j\right)=x_{c}\left(i,j\right)\times g_{c}^{h}\left(i\right)\times g_{c}^{w}\left(j\right)$$
where $y_{c}\left(i,j\right)$ represents the final output feature; $x_{c}\left(i,j\right)$ represents the original feature; $g_{c}^{h}\left(i\right)$ and $g_{c}^{w}\left(j\right)$ represent the features after aggregation processing to restore the original number of channels in the height and width dimensions respectively.

#### 2.2.3. MLLAttention for Cow Behavior Recognition

Mamba is a state-space model with linear computational complexity. Mamba can be seen as a variant of the Linear Attention Transformer with special linear attention and improved Block design. Compared with the traditional Linear Attention paradigm, Mamba has 6 different designs: input gate, forget gate, shortcut, no attention normalization, single-head, and modified block design. It is empirically verified that the forget gate and block design are largely the key to Mamba’s superior performance. In addition, it is proved that the cyclic calculation of the forget gate may not be an ideal choice for visual models. Instead, appropriate position encoding can serve as a forget gate in visual tasks while maintaining parallelized computation and fast reasoning [21].

The linear attention Transformer model usually adopts the design in Figure 9a, which consists of a linear attention module and an MLP module. In contrast, Mamba improves upon its design by combining H3 [22] and Gated Attention [23], resulting in the architecture shown in Figure 9b. The MLLAttention’s improved Mamba block integrates multiple operations, as illustrated in Figure 9c, such as selective SSM, depthwise convolution, linear mapping, activation function, gating mechanisms, etc., and is often more effective than a traditional Transformer design.

In view of the multi-scale and multi-behavior characteristics of dairy cows in intensive farming environments, based on the YOLOv8n model, the Neck is improved and the MLLAttention attention is introduced to propose CAMLLA-YOLOv8n, whose structure is shown in Figure 10. The Head of YOLOv8n adopts the Anchor-Free idea to directly predict the center point and size of the target object and no longer relies on the preset Anchor. P5 is a small feature map used to detect large targets. The MLLAttention module is first added to the deepest P5 layer. In this layer, the feature map of P5 directly enters the MLLAttention module for processing after a series of convolution layers. After the MLLAttention processing of the P5 layer, the feature map is increased in resolution by upsampling and then concatenated with the original feature map of the P4 layer. The fused feature map of the P4 layer is convolved again and input into the MLLAttention module. P3 is a large feature map used to detect small targets. After the MLLAttention processing of the P4 layer, the feature map is upsampled and fused with the original feature map of the P3 layer. After processing, it enters the MLLAttention module of the P3 layer.

By applying MLLAttention at multiple levels (P3, P4, P5), CAMLLA-YOLOv8n can not only strengthen the attention to the behavioral characteristics of cows at different scales but also improve the accuracy of behavior recognition.

Scale adaptability of behavioral characteristics: MLLAttention ensures that the model can adapt to visual information of various scales by fine-tuning feature maps from different levels. At the P5 level, by strengthening the attention to larger areas in the scene, the model can effectively identify behaviors that require a wider field of view, such as fighting and mating. At the P3 level, by strengthening the capture of local details, the ability to capture detailed activities such as licking that occur within a smaller field of view is improved.

Spatial context enhancement of behavior recognition: The behavior recognition of cows depends not only on the posture and activities of cows but also on the interaction with their surroundings. MLLAttention enhances the model’s understanding of the environmental context by integrating spatial information from different scales, thereby improving the recognition accuracy of behaviors such as standing and lying that are related to changes in environmental position.

Temporal feature analysis of dynamic behaviors: When identifying dynamic and complex behaviors such as mating or fighting, MLLAttention focuses on the changes in consecutive frames in the time series and captures the temporal dimension information in the video sequence, which is crucial for understanding the cow behavior in the video.

#### 2.2.4. SPPF-GPE for Cow Behavior Recognition

In the multi-scale challenge of cow behavior data, target sizes are widely distributed in different pixel areas (small targets are between 20 and 50 pixels wide and between 30 and 100 pixels high; medium-sized targets are between 150 and 300 pixels wide, with a height between 200 and 500 pixels; and the large-size target is between 300 and 600 pixels wide with a height between 400 and 700 pixels), which significantly affects the recognition efficiency of the model. In particular, the recognition difficulty increases due to the loss of details for small objects in deep feature maps and the larger receptive field. In order to deal with this problem, as shown in Figure 11, we have made innovative improvements to the SPPF module of YOLOv8n to form SPPF-GPE (Global Pooling Enhanced SPPF), introducing global average pooling and global maximum pooling layers, among which GAP (Global Average Pooling) provides a form of global feature representation by calculating the average of the entire feature map, which helps capture background and environmental information. GMP (Global Max Pooling) emphasizes the most significant features by extracting the maximum value in each feature map, which is associated with the key parts of the cow or behavior, highlighting the main objects or behaviors in the image, and is especially suitable for dynamic and complex behaviors. Such as fighting and mating detection. After the fusion of the global information of GAP and GMP, broader background information and highlighted key visual features are obtained. In addition, multi-scale pooling is able to capture image details from coarse to fine, which is crucial for maintaining the accuracy of action recognition under varying viewing angles and distances.

#### 2.2.5. Shape–IoU

The YOLO series of networks mainly calculates losses based on IoU loss. As shown in Figure 12, A and B are the Ground Truth Box and Prediction Box, respectively. IoU (Intersection over Union) [24] is the ratio of the intersection over the union of the Ground Truth Box and the Prediction Box, which is used to determine the degree of overlap between the Ground Truth Box and the Prediction Box. It is used to evaluate the matching degree between different Prediction Boxes and the real box in the target detection task. A high IoU value indicates that the Prediction Box has a higher overlap rate with the real box, that is, a more accurate prediction.

In this study, the above problem existed: In order to overcome the shortcomings of existing research, based on the reality that the shape and scale factors of the bounding box itself will affect the regression results, this paper uses the bounding box regression method Shape–IoU [25], which focuses on the shape and scale of the bounding box itself. By focusing on the shape and scale of the bounding box itself, the loss can be calculated, making the bounding box regression more accurate, effectively improving the detection accuracy and outperforming existing methods.

As shown in Figure 13, the scale of the Ground Truth Box in bounding box regression sample A and B is the same, while the scale of the Ground Truth Box in C and D is the same. The shape of the Ground Truth Box in A and C is the same, while the shape of the Ground Truth Box in B and D is the same. The scale of the bounding boxes in C and D is greater than the scale of the bounding boxes in A and B, and Figure 13a has the same deviation, with a shape deviation of 0. The difference between Figure 13a,b is that the shape deviation of all bounding box regression samples in Figure 13b is the same, with a deviation of 0.

The following conclusions can be drawn:

The impact of non-square boxes: If the Ground Truth Box is not square and has sides of different lengths when the deviation and shape deviation are the same and not all 0, then the difference in shape and size of the bounding box will lead to changes in its IoU value.

The impact of different shapes at the same scale: For bounding boxes with the same scale but different shapes, the difference in shape will affect the IoU value. Especially when the deviation is along the short side direction, the change in the IoU value is more significant.

Sensitivity of small-scale boxes: The IoU values of the smaller-scale bounding box regression samples with the same shape are more susceptible to the influence of the shape of the Ground Truth Box than the Prediction Boxes of larger sizes.

The formula for Shape–IOU can be derived from Figure 14.

IoU Metric: The IoU, which is the most popular criterion for evaluating target detection, is defined as follows:(8)$$\mathrm{IoU}=\frac{|B\;\cap \;B^{\mathrm{gt}}|}{|B\;\cup \;B^{\mathrm{gt}}|}$$
where B and $B^{\mathrm{gt}}$ represent the Predicted Box and the Ground Truth Box, respectively.
(9)$$\mathrm{ww}=\frac{2\times {\left(w^{\mathrm{gt}}\right)}^{scale}}{{\left(w^{\mathrm{gt}}\right)}^{scale}+{\left(h^{\mathrm{gt}}\right)}^{scale}}$$
(10)$$\mathrm{hh}=\frac{2\times {\left(h^{\mathrm{gt}}\right)}^{scale}}{{\left(w^{\mathrm{gt}}\right)}^{scale}+{\left(h^{\mathrm{gt}}\right)}^{scale}}$$
where “scale” is a scaling factor related to the size of the targets in the dataset, and “ww” and “hh” are the weighting coefficients for the horizontal and vertical directions, respectively, which are determined by the shape of the Ground Truth Box.
(11)$${\mathrm{distance}}^{\mathrm{shape}}=\mathrm{hh}\times {\left(x_{c}-x_{c}^{\mathrm{gt}}\right)}^{2}/c^{2}+\mathrm{ww}\times {\left(y_{c}-y_{c}^{\mathrm{gt}}\right)}^{2}/c^{2}$$
is the weighted Euclidean distance between the center points of the Prediction Box and the Ground Truth Box, where “hh” acts as a weighting factor for the height affecting the distance calculation, and “c” is a normalization constant that adjusts the impact of center point shifts.
(12)$$\Omega^{shape}=\sum_{t=w,h}{\left(1-e^{-\omega_{t}}\right)}^{\theta },\theta =4$$

Weighted summation is performed for the shape discrepancies at each time step, where $\omega_{t}$ represents the shape error measure at each time step, and θ is a weighting parameter used to enhance or mitigate the impact of errors.
(13)$$\left\{\begin{array}{c} \omega_{w}=\mathrm{hh}\times \frac{\left|w-w^{\mathrm{gt}}\right|}{\max\left(w,w^{\mathrm{gt}}\right)}\\ \omega_{h}=\mathrm{ww}\times \frac{\left|h-h^{\mathrm{gt}}\right|}{\max\left(h,h^{\mathrm{gt}}\right)} \end{array}\right.$$

The above equations calculate the relative values of width and height errors. Shape–IoU and its corresponding bounding box regression loss combine IoU, ${\mathrm{distance}}^{\mathrm{shape}}$, and $\Omega^{\mathrm{shape}}$, and the formula is as follows:(14)$$L_{\mathrm{Shape}\text{-}\mathrm{IoU}}=1-\mathrm{IoU}+{\mathrm{distance}}^{\mathrm{shape}}+0.5\times \Omega^{\mathrm{shape}}$$

The weighting factor 0.5 is used to balance the contribution of shape error, ensuring that the model does not sacrifice too much position and size accuracy while maintaining shape recognition. The original loss function of YOLOv8n uses CIoU loss as the bounding box regression loss function. The calculation formula of CIoU is as follows:(15)$$L_{CIoU}=1-IoU+\frac{\rho^{2}\left(b,b^{gt}\right)}{c^{2}}+\alpha \nu$$

The CIoU introduces the aspect ratio between the predicted box and the Ground Truth Box, making it more attentive to the shape of the bounding box. However, it also increases computational complexity and makes parameter tuning more challenging. Therefore, this experiment uses Shape–IoU to replace the original CIoU loss, which can calculate the loss by focusing on the shape and scale of the bounding box itself, thereby making the bounding box regression more accurate.

## 3. Results

### 3.1. Evaluation Indicators

In order to verify the effectiveness of the proposed CAMLLA-YOLOv8n model, in this study, the network performance is evaluated by six evaluation indicators: Precision, Recall, mAP@0.5, mAP@0.5:0.95, Params, and FLOPs. The calculation of Precision and Recall is shown in Equations (Equation 16) and (Equation 17).
(16)$$\mathrm{Precision}=\frac{\mathrm{TP}}{\left(\mathrm{TP}+\mathrm{FP}\right)}$$
(17)$$\mathrm{Recall}=\frac{\mathrm{TP}}{\left(\mathrm{TP}+\mathrm{FN}\right)}$$

Among them, TP, FP, FN and TN are the number of actual positive classes predicted as positive, the number of actual negative classes predicted as positive, the number of actual positive classes predicted as negative and the number of actual negative classes predicted as negative, respectively. Precision indicates the proportion of true positive samples predicted as positive. Recall indicates the proportion of all actual positive samples correctly identified as positive. mAP@0.5 means that when IoU is set to 0.5, the AP value of all images of each category is calculated, and then all categories are averaged; mAP@0.5:0.95 means the average mAP at different IoU thresholds (from 0.5 to 0.95, step size 0.05). The higher the value, the better the detection effect of the target detection model. mAP is a key indicator to measure the quality of the target detection algorithm. It is calculated by combining the Precision and Recall and the area under the P-R curve. Params, which is used to measure the memory usage of the model, is the sum of all trainable parameters in the network. FLOP refers to the number of floating-point operations performed during a model’s inference process and is an important indicator for measuring model complexity and computational efficiency.

### 3.2. Experimental Environment Configuration and Hyperparameter Settings

To ensure a fair comparison, all network models in the experiment are implemented and executed in the Pytorch framework. The hardware configuration used in this study is as follows: the operating system is Ubuntu 20.04, the CPU model is Intel(R) Xeon(R) Platinum 8352V CPU @ 2.10 GHz, the GPU model is NVIDIA GeForce RTX 4090, the GPU memory is 24 GIB, the RAM is 64 GB DDR5 (32 × 2 GB), and the solid-state drive capacity is 4 TB. The deep learning framework used is Pytorch 2.2.2, the CUDA version is 12.2, the cuDNN version is 8.7, and the Python version is 3.9.19. The details of the experimental environment settings are shown in Table 3.

In the training of deep learning models, the setting of hyperparameters has a significant impact on the learning efficiency and final performance of the model. During the model training process, the experimental parameters were set as follows: the input image size was set to 640 × 640 pixels, and the batch size was 64; to speed up the convergence speed, the initial learning rate was set to 0.01, the attenuation coefficient was set to 0.0005, and the momentum factor was set to 0.937. BatchSize was set to 50 epochs, and the Adam optimizer was used to iteratively optimize network parameters. The other parameters were YOLOv8 official default parameters. After confirming model convergence, we saved the model parameters and evaluated the model.

### 3.3. Comparative Analysis of Different Models of Cow Behavior Recognition

The performance of deep learning models needs to be compared between models. In order to evaluate the effectiveness of the CAMLLA-YOLOv8n model in cow behavior recognition and further analyze the performance of the CAMLLA-YOLOv8n model, the same dataset as the CAMLLA-YOLOv8n model is used to evaluate the performance of six models, including YOLOv3 [26], YOLOv5n, YOLOv5s, YOLOv7tiny [27], YOLOv8n and YOLOv8s. From the experimental results in Table 4, it can be seen that under the same experimental conditions, compared with the YOLOv8n model, the CAMLLA-YOLOv8n cow behavior recognition algorithm proposed by us has achieved 2.18%, 1.62%, 1.84% and 1.77% improvement in Precision, Recall, mAP@0.5 and mAP@0.5:0.95, respectively. In terms of Params and FLOPs, the Params of the CAMLLA-YOLOv8n model are 0.242 M larger than those of YOLOv8n, which is basically unchanged. The FLOPs of the CAMLLA-YOLOv8n model increased slightly, which improved the accuracy of cow behavior recognition to a certain extent. Although the lightweight CAMLLA-YOLOv8n is second only to YOLOv5n, the detection accuracy of YOLOv5n lags far behind that of CAMLLA-YOLOv8n.

The comprehensive performance comparison of different detection algorithms is shown in Figure 15. The distance between each curve and each axis represents the performance of the algorithm on the corresponding indicator. The larger the area surrounded by the curve, the better the overall performance of the algorithm. The closer the intersection of each curve with each axis is to the outermost circle, the better the performance of the indicator is, and the closer it is to the innermost circle, the worse the performance of the indicator is. YOLOv5n (orange curve) is close to the outermost circle in terms of Params and FLOPs, indicating that the YOLOv5n model is relatively lightweight in terms of Params and FLOPs. The CAMLLA-YOLOv8n model’s Params and FLOPs are second only to YOLOv5n and YOLOv8n. Despite its lower resource consumption, its performance is the highest. Overall, the area surrounded by CAMLLA-YOLOv8n (red curve) in the figure is the largest, indicating that the CAMLLA-YOLOv8n model has achieved the highest level in the four key indicators of Precision, Recall, mAP@0.5, and mAP@0.5:0.95, and has the best comprehensive performance among all current models.

Figure 16 shows the dynamic changes in the four key performance indicators of Precision, Recall, mAP@0.5 and mAP@0.5:0.95 of seven different YOLO models during training. In the initial stage of model training, all model performance indicators increased rapidly, and as the training progressed, these performance indicators gradually stabilized. Specifically, the CAMLLA-YOLOv8n model demonstrates high stability and excellent performance in four key metrics: Precision, Recall, mAP@0.5, and mAP@0.5:0.95. Both Precision and Recall stabilize after the 30th epoch, with a slight trend of performance improvement in later stages, where fluctuations in Precision and Recall remain around 94%.

Figure 17 depicts the changes in the loss of seven different YOLO models during the training and validation stages, including Box Loss, DFL Loss and CLS Loss. In the initial stage of model training, due to the high learning rate, the loss curve dropped rapidly within the first 10 epochs, indicating that the model’s adaptability to the training data was enhanced. The CAMLLA-YOLOv8n model performs particularly well in terms of validation loss. As the training progresses, its loss curve gradually stabilizes and basically converges after about 30 epochs, with the loss value fluctuating around 0.3. This shows that while ensuring high-precision detection, CAMLLA-YOLOv8n also has strong generalization capabilities and can effectively suppress overfitting.

## 4. Discussion

### 4.1. Impact of Improved Modules on Algorithms

The proposed CAMLLA-YOLOv8n model is based on YOLOv8n. It introduces CA attention in the C2f module and the attention mechanism in the P3, P4, and P5 layers of Neck; innovatively improves the SPPF module to form the SPPF-GPE module; and replaces the IoU with the Shape–IoU loss function. In order to evaluate the impact of each optimization module of CAMLLA-YOLOv8n on the model performance, the variable control method is used for ablation experiments. Training and validation are performed on the same dataset and parameters, and the experimental results are shown in Table 5.

In order to deal with the challenges of complex background and increased number of cows caused by Mosaic fusion, it is necessary to more accurately identify and understand the spatial relationship between different cow positions and enhance the model’s sensitivity to key areas and filtering ability of background interference. First, after integrating the Coordinate Attention mechanism in the C2f module to form the C2f-CA module, the model’s mAP@0.5 and mAP@0.5:0.95 increased by 0.46% and 0.11%, respectively. Considering the challenges of cow behavior recognition due to large-scale changes, the MLLAttention mechanism was introduced in the P3, P4, and P5 layers of Neck on the basis of YOLOv8n+CA. At this time, the Precision and Recall of the model were improved by 0.85% and 0.31%, respectively. Secondly, the SPPF module is innovatively formed into the SPPF-GPE module, which optimizes small target recognition by combining global average pooling and global maximum pooling. The Precision and Recall of the model were improved by 1.19% and 0.69%, respectively. Finally, Shape–IoU loss optimization was introduced to focus on the shape and scale features of the Bounding Box and improve the matching degree between the Prediction Box and the Ground Truth Box. The Precision and Recall of CAMLLA-YOLOv8n were increased by 2.18% and 1.62%, respectively.

The comprehensive experimental results show that each optimization module in the CAMLLA-YOLOv8n model improved the recognition accuracy to varying degrees, proving the effectiveness of each optimization operation. Figure 18 shows the performance comparison of the cow behavior recognition model under different optimization strategies. Compared with the YOLOv8n model, although the Params and FLOPs of the CAMLLA-YOLOv8n model increased slightly, it achieved significant improvements of 2.18%, 1.62%, 1.84% and 1.77% in the four key performance indicators of Precision, Recall, mAP@0.5 and mAP@0.5:0.95, respectively.

### 4.2. Heat Map Visualization Analysis

To investigate the improvements in the CAMLLA-YOLOv8n model and visually present the comparison before and after optimization, we used GradCAM to perform a visual analysis of the key region responses of both YOLOv8n and the optimized CAMLLA-YOLOv8n model. As shown in Figure 19, the blue area represents the part with lower confidence, while the red area represents the part with higher confidence. It can be seen that compared to YOLOv8n, CAMLLA-YOLOv8n is able to focus more intensively on key feature regions and heatmap values in dense areas under the same scenario, covering a larger number of target areas that need to be detected. At the same time, when focusing on a broader range of features, CAMLLA-YOLOv8n effectively suppresses interference from irrelevant information, improving the accuracy of cow behavior recognition. YOLOv8n pays attention to some irrelevant feature information while paying attention to a wider range of features.

### 4.3. CAMLLA-YOLOv8n Behavior Recognition Detection Results and Visual Analysis

In order to further verify the accuracy and detection effect of the CAMLLA-YOLOv8n model in the cow behavior recognition task, this study randomly selected three representative images from the test set for testing to demonstrate the model’s performance in various scenarios. Figure 20 shows the test results. The first line is the original image, the second line is the manually annotated Ground Truth Box, the third line shows the detection results of YOLOv8n, and the fourth line is the improved CAMLLA-YOLOv8n detection results. By comparison, CAMLLA-YOLOv8n shows significant advantages in handling occlusions between cows, dealing with multi-scale targets, and adapting to complex backgrounds. Specifically, in the first scene, due to mutual occlusion between cows and the limitations of the monitoring range, the traditional YOLOv8n model struggles to identify partially occluded cows. For example, the partially exposed standing cow in the upper left is not detected, resulting in missed detection. Simultaneously, YOLOv8n’s detection accuracy for cows grazing on the lower left side is only 0.71, while CAMLLA-YOLOv8n’s detection accuracy reaches 0.91, demonstrating its superiority in handling scenes with occlusions and complex backgrounds. In the second scene, both models have high recognition accuracy for the cow standing in the middle, but YOLOv8n’s accuracy is 0.95, which is lower than CAMLLA-YOLOv8n’s 0.97. In the third scene, CAMLLA-YOLOv8n’s accuracy in identifying licking behavior reaches a perfect 1.00, slightly higher than YOLOv8n’s 0.99. For cows lying down in the distance, although the color of the ground and the fence may interfere with the detection, CAMLLA-YOLOv8n’s detection accuracy is still generally higher than YOLOv8n’s by about 0.02. In summary, the CAMLLA-YOLOv8n model proposed in this article shows better performance than the traditional model in terms of the diversity of cow behaviors, occlusion processing, and adaptability to background complexity, effectively improving the accuracy of cow behavior recognition.

In terms of real-time detection, we first tested the system on an NVIDIA GeForce GTX 1080 Ti with the following configuration: Windows 10 operating system, AMD R7-5700X CPU (3.4 GHz), NVIDIA GeForce GTX 1080 Ti GPU, 11 GB of video memory, 32 GB DDR4 RAM (16 × 2 GB), and a 1TB solid-state drive. The average inference time of the CAMLLA-YOLOv8n model for processing a single image is 35.5 ms, and the inference speed meets the requirements for real-time detection of video streams at 30 fps. In addition, considering the application scenarios of edge computing devices, we also deployed and tested the model on the NVIDIA Jetson AGX Xavier edge development board and the HUAWEI Atlas 200 DK A2 development kit. On these edge devices, the CAMLLA-YOLOv8n model also achieved real-time processing speed, which demonstrates its good adaptability and reliability on edge devices of different performance levels.

During the training process of the CAMLLA-YOLOv8n model, the visualization results generated on dataset images, as shown in Figure 21 and Figure 22, demonstrate that CAMLLA-YOLOv8n performs well in various challenging scenarios. It accurately identifies the behavior of cows in different environments, addressing the background complexity caused by Mosaic fusion and the increase in the number of cows, as well as the multi-scale, multi-behavioral characteristics and small targets. This effectively completes the recognition task of this study.

## 5. Conclusions

Existing recognition methods are not ideal for cow behavior data recognition. Therefore, this paper proposes an improved cow behavior algorithm CAMLLA-YOLOv8n based on YOLOv8n. Due to the complexity of the background caused by the Mosaic fusion of the dataset, the increase in the number of cows, the large changes in target scale, the large number of small target samples, and the diverse cow behaviors in the real group farming environment, the accuracy of behavior recognition and the model detection results are easily disturbed. Based on YOLOv8n, the detection accuracy of the model is improved by using three network structures, C2f-CA, MLLAttention, and SPPF-GPE, combined with Shape–IoU loss. The main contributions of this paper are as follows: In order to solve the challenges of the complex background and the increased number of cows caused by the Mosaic fusion of the dataset, the Coordinate Attention mechanism is integrated into the C2f module to form the C2f-CA module, which strengthens the expression of inter-channel feature information so that the model can more accurately identify and understand the spatial relationship between different cow positions, and improve the model’s sensitivity to key areas and the ability to filter background interference. Secondly, the MLLAttention mechanism is introduced in the P3, P4, and P5 layers of the Neck model to cope with the challenges of cow behavior recognition caused by large-scale changes. In addition, we also innovatively improved the SPPF module to form the SPPF-GPE module, which optimizes small target recognition and enhances the model’s ability to respond to environmental changes and capture key parts of cow behavior by combining global average pooling and global maximum pooling. Finally, in view of the limitations of traditional IoU loss in cow behavior detection, we replaced CIoU loss with Shape–IoU loss, focusing on the shape and scale features of the Bounding Box, thereby improving the matching degree between the Prediction Box and the Ground Truth Box.

Finally, although the Params and FLOPs of the CAMLLA-YOLOv8n model are slightly increased compared to the YOLOv8n model, it achieved significant improvements of 2.18%, 1.62%, 1.84%, and 1.77% in the four key performance indicators of Precision, Recall, mAP@0.5, and mAP@0.5:0.95, respectively. This indicates that the CAMLLA-YOLOv8n model can effectively meet the needs of accurate and fast recognition of cow behavior in practical application scenarios. In the future, we will use model compression methods such as pruning and quantization to reduce the number of model parameters and better deploy them on edge devices while maintaining a balance between speed and accuracy.

## Figures and Tables

**Figure 1 animals-14-03033-f001:**
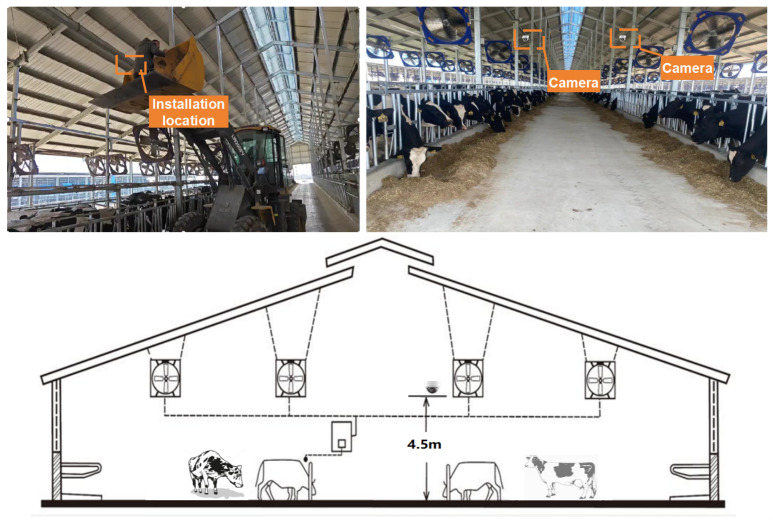
Camera installation location diagram.

**Figure 2 animals-14-03033-f002:**
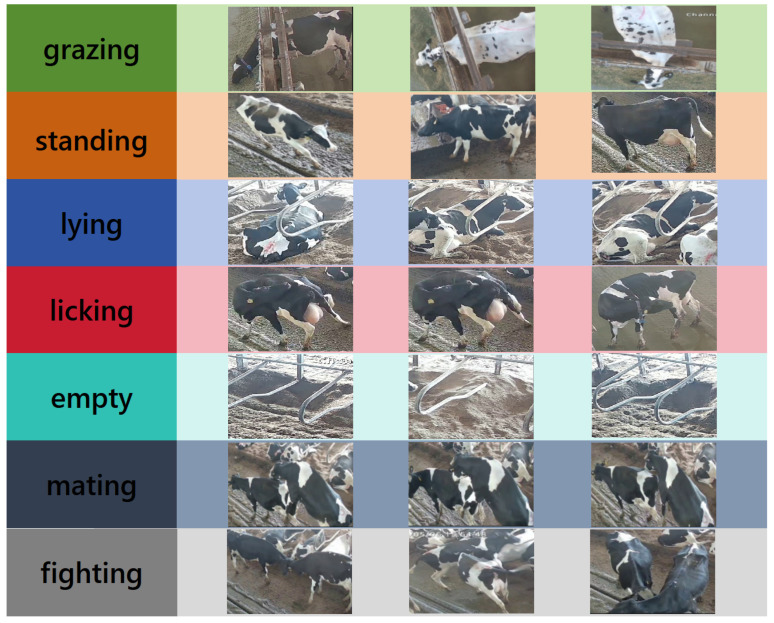
Example of sample images of cow behavior in the dataset.

**Figure 3 animals-14-03033-f003:**
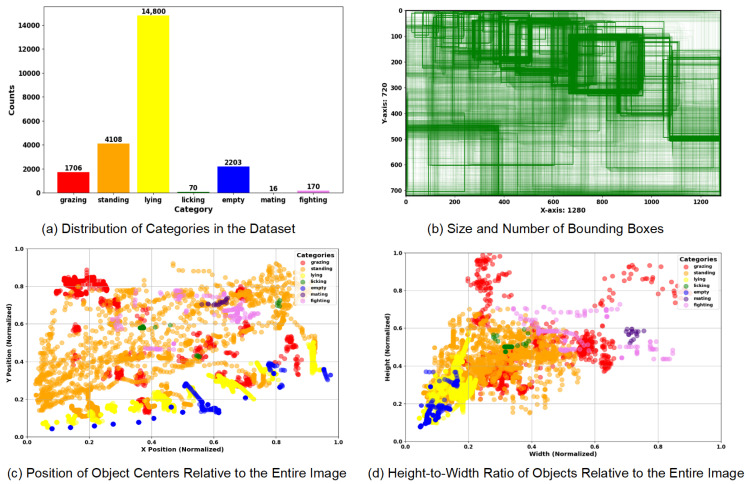
Category distribution and annotation information of the dataset.

**Figure 4 animals-14-03033-f004:**
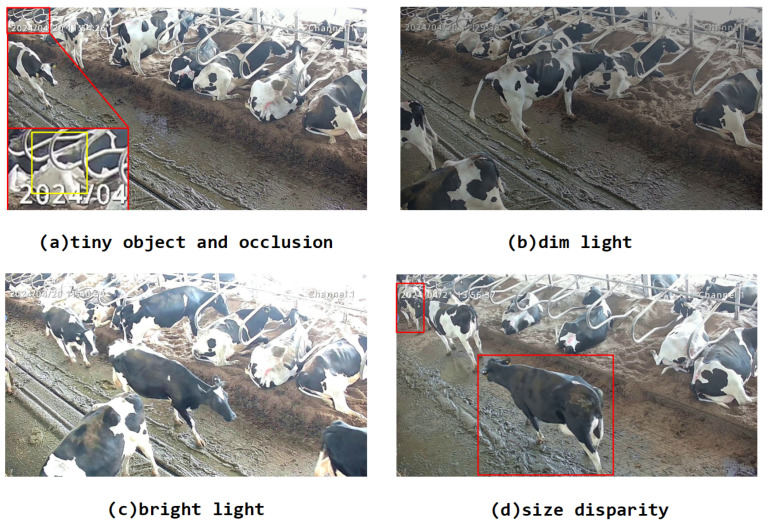
Sample images from the challenging dataset.

**Figure 5 animals-14-03033-f005:**
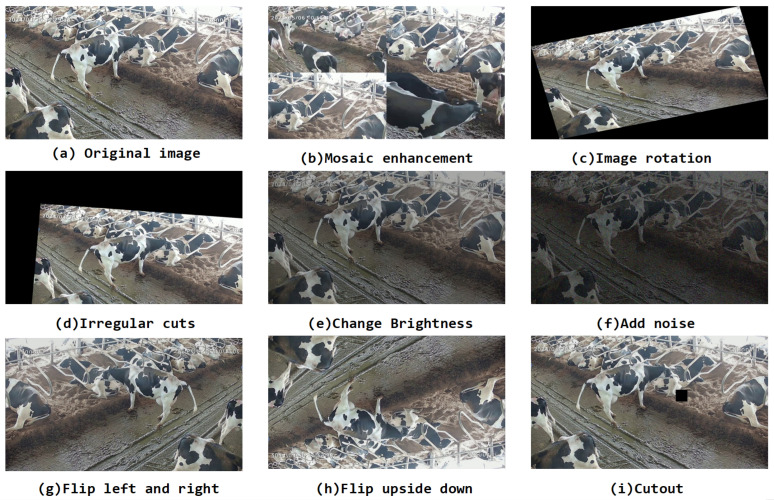
Example of augmented data.

**Figure 6 animals-14-03033-f006:**
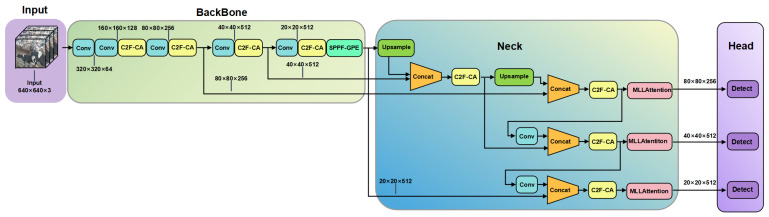
CAMLLA-YOLOv8n cow behavior recognition network architecture diagram.

**Figure 7 animals-14-03033-f007:**
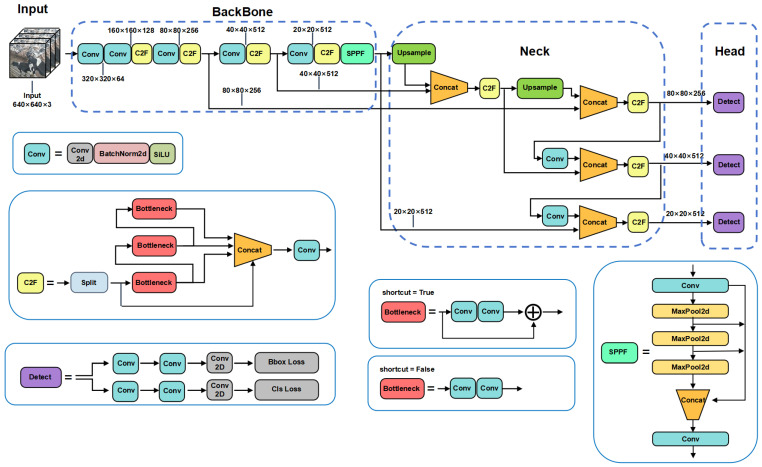
YOLOv8n overall network structure diagram.

**Figure 8 animals-14-03033-f008:**
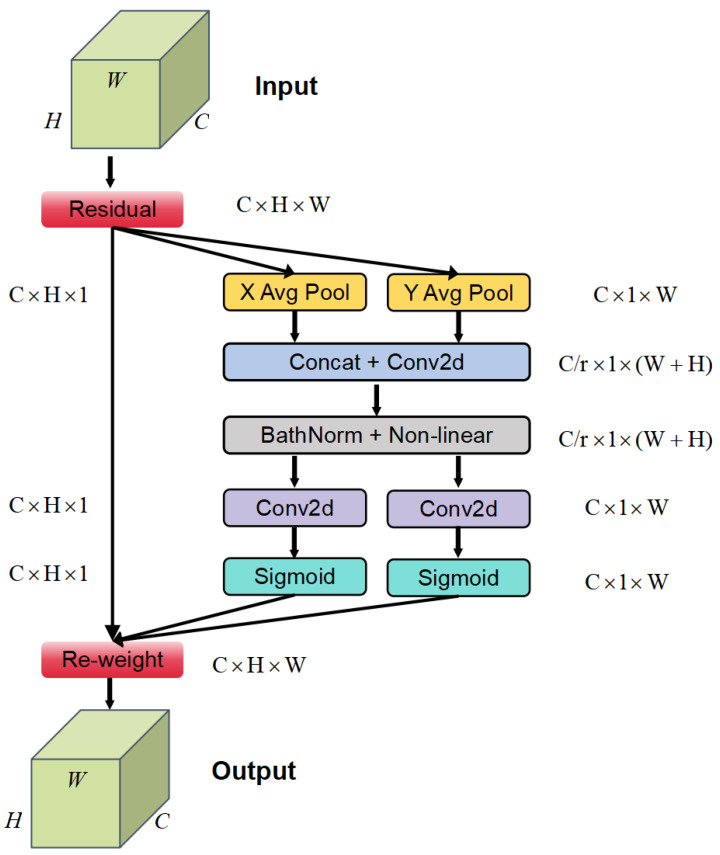
CA attention module diagram.

**Figure 9 animals-14-03033-f009:**
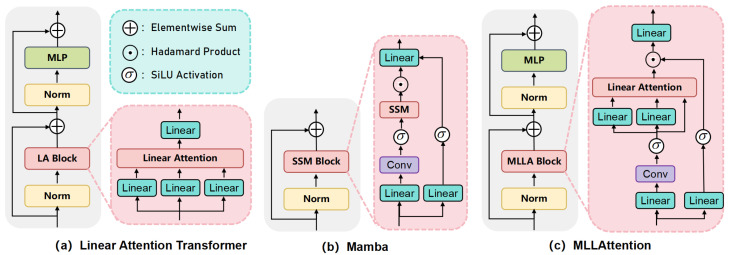
Linear Attention Transformer architecture, Mamba architecture, and MLLAttention architecture.

**Figure 10 animals-14-03033-f010:**
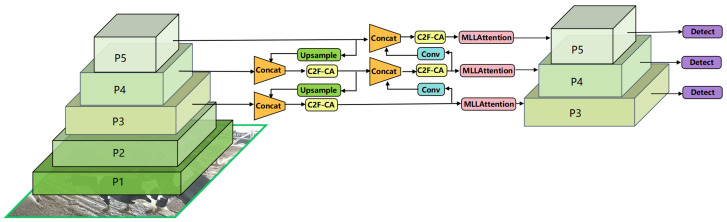
Multi-level feature fusion and MLLAttention Mechanisms display of CAMLLA-YOLOv8n backbone network.

**Figure 11 animals-14-03033-f011:**
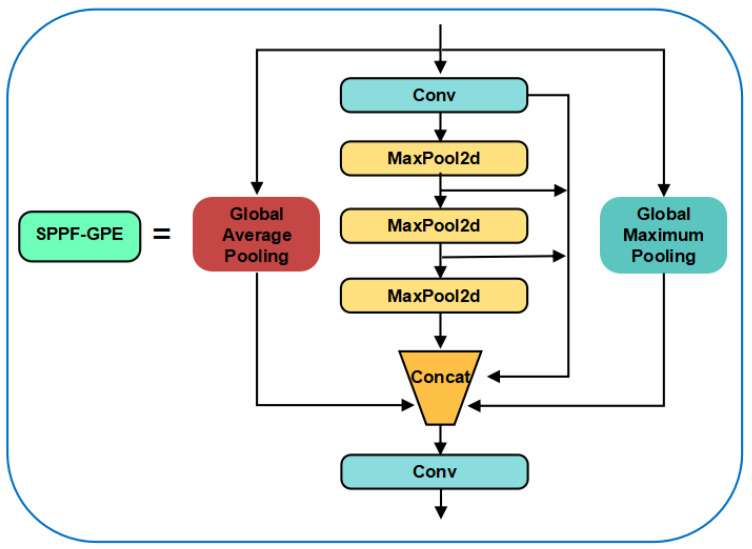
SPPF-GPE structure diagram.

**Figure 12 animals-14-03033-f012:**
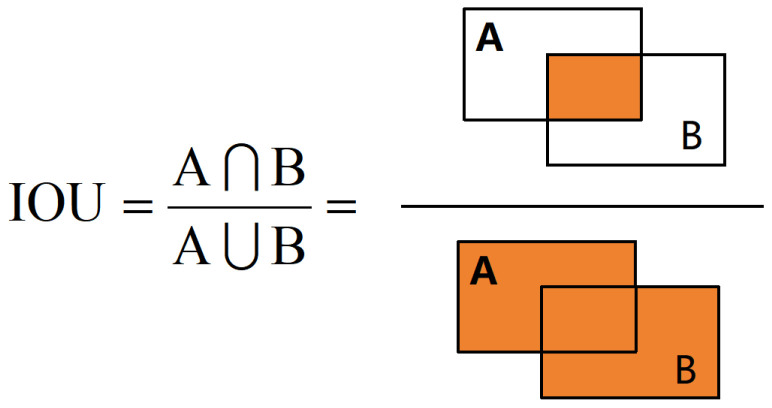
IOU calculation formula diagram. A represents the ground truth bounding box, and B represents the predicted bounding box.

**Figure 13 animals-14-03033-f013:**
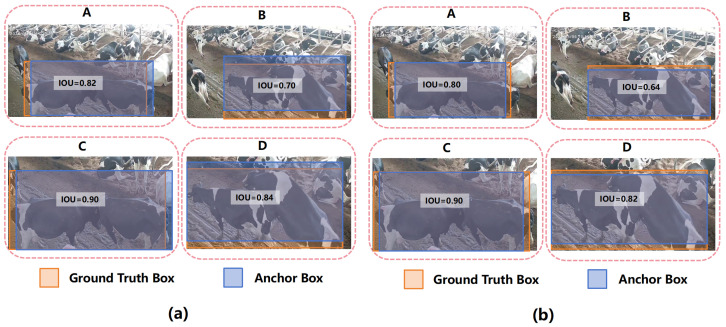
IOU comparisons for Anchor and Ground Truth Boxes. (**a**) Shows boxes with the same shape deviation but different scales. (**b**) Shows boxes with the same shape and scale, all with a shape deviation of 0.

**Figure 14 animals-14-03033-f014:**
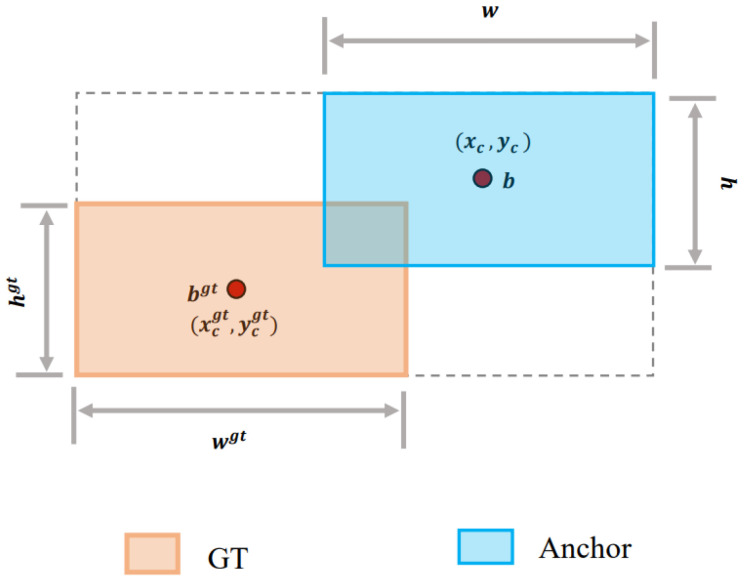
Schematic diagram of Ground Truth Box and Anchor Box.

**Figure 15 animals-14-03033-f015:**
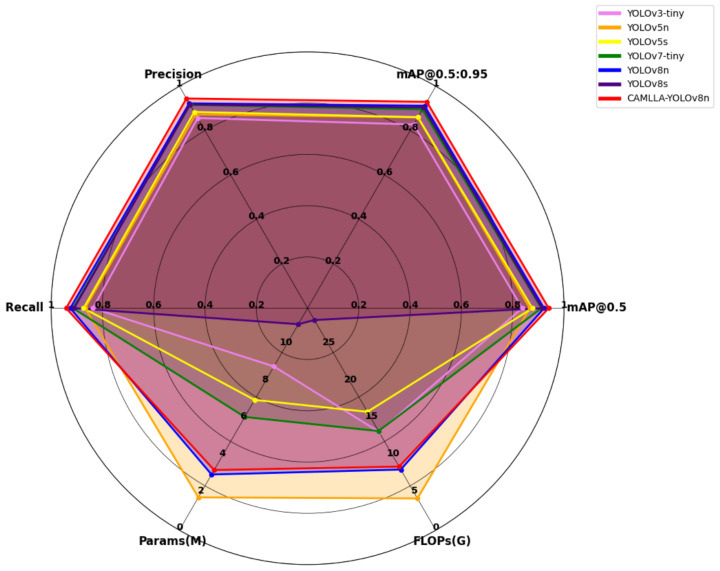
Comprehensive performance comparison of seven YOLO detection algorithms.

**Figure 16 animals-14-03033-f016:**
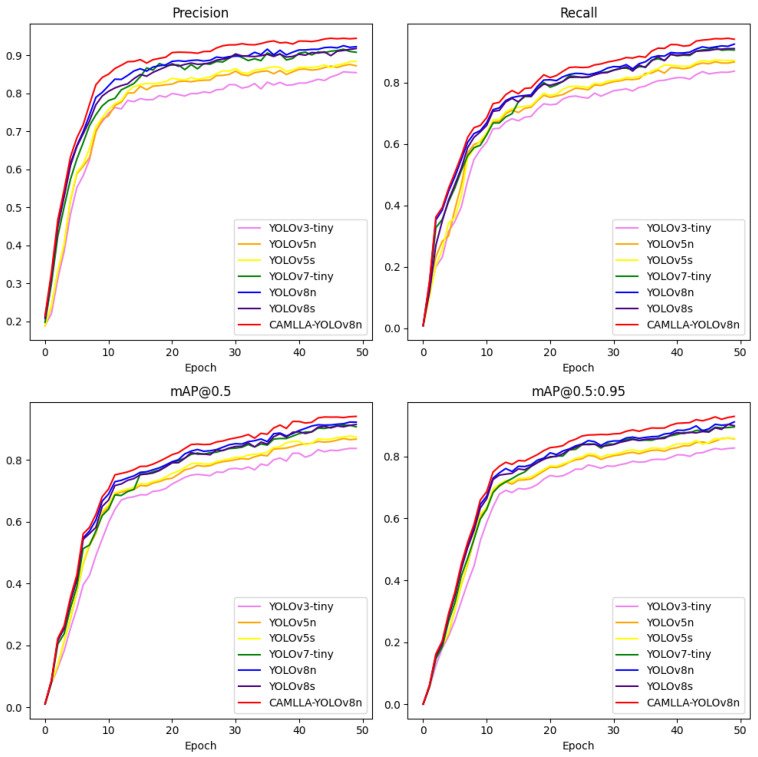
Comparative analysis of Precision, Recall, and Mean Average Precision across seven YOLO detection algorithms.

**Figure 17 animals-14-03033-f017:**
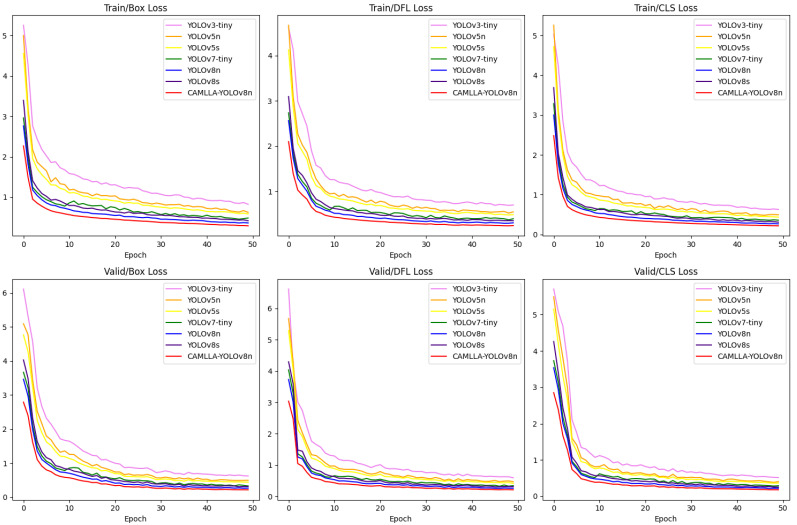
Training and validation loss profiles across seven YOLO detection algorithms.

**Figure 18 animals-14-03033-f018:**
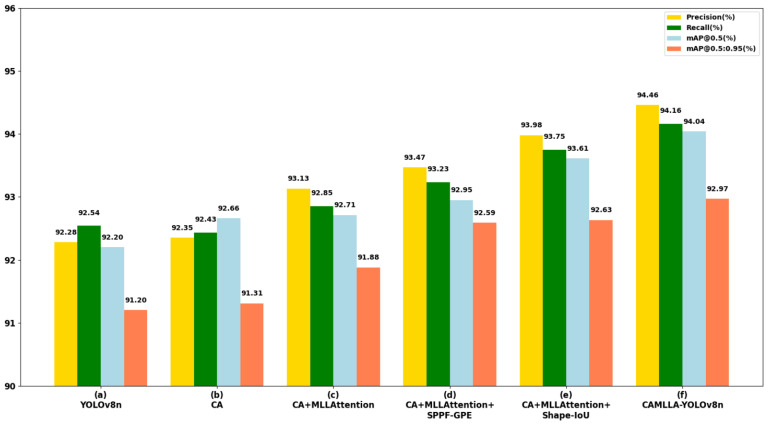
Visualization of the ablation results of different optimization modules on Precision, Recall, mAP@0.5, and mAP@0.5:0.95.

**Figure 19 animals-14-03033-f019:**
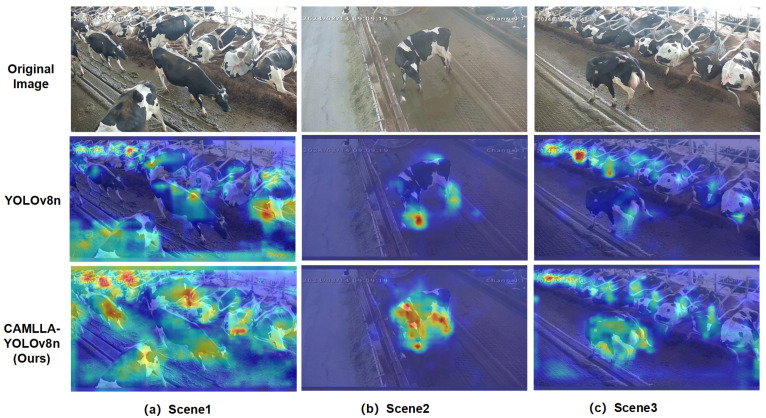
Heatmap comparison between YOLOv8n and CAMLLA-YOLOv8n. Note: The comparison is shown in three scenarios. The first row is the original image, the second row is the YOLOv8n heatmap, and the third row is the optimized CAMLLA-YOLOv8n heatmap.

**Figure 20 animals-14-03033-f020:**
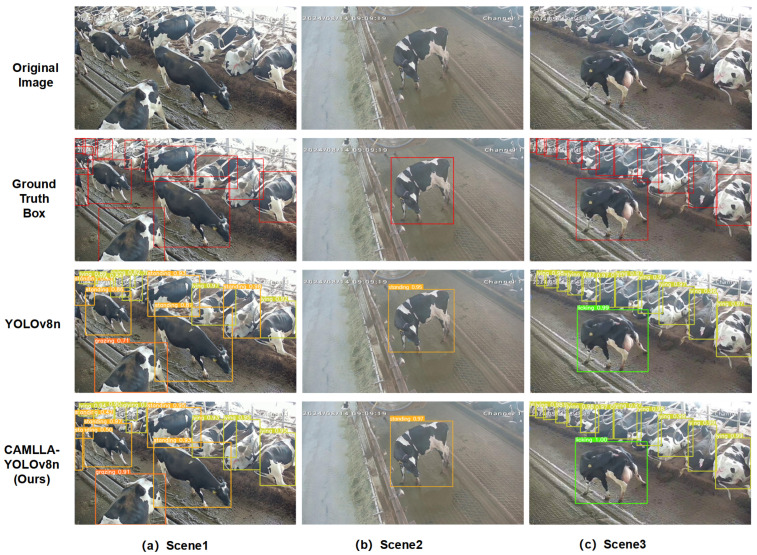
Comparison of test results between YOLOv8n and CAMLLA-YOLOv8n. Note: The first row shows the original image, the second row shows the manually annotated Ground Truth Box, the third row shows the detection results of YOLOv8n, and the fourth row shows the improved detection results of CAMLLA-YOLOv8n.

**Figure 21 animals-14-03033-f021:**
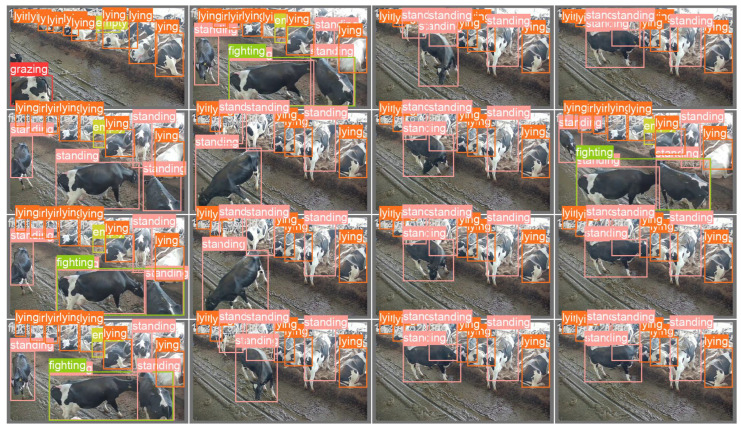
Visualization of CAMLLA-YOLOv8n detection results 1.

**Figure 22 animals-14-03033-f022:**
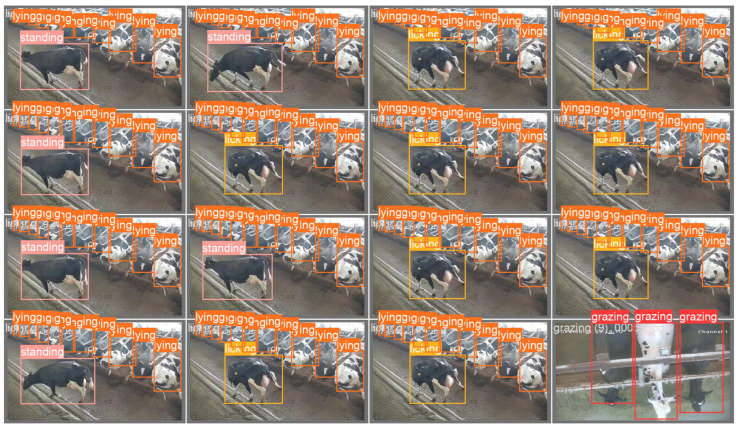
Visualization of CAMLLA-YOLOv8n detection results 2.

**Table 1 animals-14-03033-t001:** The specific parameters of our cow behavior dataset.

Item	Parameter
Number of categories	7
Category Name	Grazing, Standing, Lying, Licking, Empty, Mating, Fighting
Video Number	60
Single Video duration	15 s–30 s
Video frame rate	30 fps
Resolution	1280 × 720

**Table 2 animals-14-03033-t002:** Cow behavior criteria.

Category	Description of Behavior	Labels
Eat	Head intersects with feed trough area.	Grazing
Stand	The cow’s legs are straight to support the body.	Standing
Lie	The cow lies down in the cow bed.	Lying
Lick	Cows lick themselves or other cows with their tongues.	Licking
Empty	The cow bed in the lying area is empty, with no cows.	Empty
Mount	Two cows climb over each other in heat.	Mating
Fight	Fights or physical confrontations between cows.	Fighting

**Table 3 animals-14-03033-t003:** Environment configuration table.

Configuration Item	Value
Operating System	Ubuntu 20.04
CPU	Intel(R) Xeon(R) Platinum 8352V
RAM	64 GB
SSD	4 TB
GPU	GeForce RTX 4090
Deep Learning Framework	Pytorch 2.2.2 + cu118
CUDA Version	12.2
cuDNN Version	8.7
Python Version	3.9.19

**Table 4 animals-14-03033-t004:** Comparisonof comprehensive performance results of seven detection algorithms.

Model	Precision	Recall	mAP@0.5	mAP@0.5:0.95	Params	FLOPs
YOLOv3-tiny	0.8567	0.8372	0.8368	0.8276	8.852 M	13.3 G
YOLOv5n	0.8730	0.8674	0.8668	0.8596	1.776 M	4.3 G
YOLOv5s	0.8840	0.8735	0.8767	0.8604	7.043 M	16.0 G
YOLOv7-tiny	0.9160	0.9146	0.9118	0.8956	6.122 M	13.4 G
YOLOv8n	0.9228	0.9254	0.9220	0.9120	3.012 M	8.2 G
YOLOv8s	0.9177	0.9108	0.9144	0.9040	11.136 M	28.4 G
CAMLLA-YOLOv8n	0.9446	0.9416	0.9404	0.9297	3.254 M	8.6 G

**Table 5 animals-14-03033-t005:** Ablation experimental results of different optimization modules.

Item	C2F-CA	MLLAttention	SPPF-GPE	Shape–IoU	Precision	Recall	mAP@0.5	mAP@0.5:0.95
(a) YOLOv8n					0.9228	0.9254	0.9220	0.9120
(b) YOLOv8n	✓				0.9235	0.9243	0.9266	0.9131
(c) YOLOv8n	✓	✓			0.9313	0.9285	0.9271	0.9188
(d) YOLOv8n	✓	✓	✓		0.9347	0.9323	0.9295	0.9259
(e) YOLOv8n	✓	✓		✓	0.9398	0.9375	0.9361	0.9263
(f) CAMLLA-YOLOv8n	✓	✓	✓	✓	0.9446	0.9416	0.9404	0.9297

## Data Availability

The data presented in this study are available on request from the corresponding author.

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
