# Peer review of "CAMLLA-YOLOv8n: Cow Behavior Recognition Based on Improved YOLOv8n"

_animals, 2024, doi:10.3390/ani14203033_

Round 1
Reviewer 1 Report
Comments and Suggestions for Authors
This paper is very interesting, and looks suitable for publication. This reviewer suggest the next minor revisions:
1. Authors should improve the quality and size of all figures, in special: Fig. 2, Fig. 3, Fig. 6, Fig. 7, Fig. 16, Fig. 17, and Fig. 18, since they are not readable.
2. Authors should add a table comparing their proposal with other methods, highlighting the percentage of improvement and accuracy in each case of study.
3. ¿Is this method applicable to detect any risk behavior in human being? ¿How could it be implemented?
4. ¿How easy is to implement it in real time?
Author Response
Reviewer 1:
This paper is very interesting, and looks suitable for publication. This reviewer suggest the next minor revisions:
1.Authors should improve the quality and size of all figures, in special: Fig. 2, Fig. 3, Fig. 6, Fig. 7, Fig. 16, Fig. 17, and Fig. 18, since they are not readable.
Response: According to the reviewers' valuable suggestions, we realize that the clarity of figures is crucial to understanding the performance of the model. Therefore, we have redesigned and optimized the quality of Figures 2, 3, 6, 7, 16, 17, and 18 to ensure that the figures have sufficient resolution and clarity in the final submitted paper so that all readers can clearly read and understand what is presented in the figures.
2.Authors should add a table comparing their proposal with other methods, highlighting the percentage of improvement and accuracy in each case of study.
Response: According to the reviewer's suggestion, in order to more clearly show the differences and advantages between CAMLLA-YOLOv8n and other existing methods in this study, we comprehensively compare the comprehensiveness of seven detection algorithms including CAMLLA-YOLOv8n in Table 4. Table 4 shows the Precision, Recall, mAP50, and mAP@0.5-0.95 indicators of our proposed model CAMLLA-YOLOv8n and YOLOv3-tiny, YOLOv5n, YOLOv5s, YOLOv7-tiny, YOLOv8n, and YOLOv8s when detecting six behaviors of cows such as eating, standing, lying, licking, estrus, and fighting, as well as the empty state of the cow bed. It also includes the number of parameters (Params) and floating-point operations (FLOPs) of the model. Such a comparison not only reflects the performance of our model in various key indicators, but also highlights its advantages and potential in practical applications. The original text is as follows:“The experimental results show that, compared with models such as YOLOv3-tiny, YOLOv5n, YOLOv5s, YOLOv7-tiny, YOLOv8n, and YOLOv8s, the improved CAMLLA-YOLOv8n model achieved increases in Precision of 8.79%, 7.16%, 6.06%, 2.86%, 2.18%, and 2.69%, respectively, when detecting the states of Holstein cows grazing, standing, lying, licking, estrus, fighting, and empty bedding. Finally, although the Params and FLOPs of the CAMLLA-YOLOv8n model increased slightly compared with the YOLOv8n model, it achieved significant improvements of 2.18%, 1.62%, 1.84%, and 1.77% in the four key performance indicators of Precision, Recall, mAP50, and mAP@0.5-0.95, respectively.”
3.Is this method applicable to detect any risk behavior in human being? ¿How could it be implemented?
Response: Thanks to the reviewer for his suggestion. Regarding whether the CAMLLA-YOLOv8n method is suitable for detecting dangerous human behaviors, we believe that the CAMLLA-YOLOv8n method can be used to detect dangerous human behaviors in principle. Although the CAMLLA-YOLOv8n model is specially designed and optimized for identifying the behavior of Holstein cows, CAMLLA-YOLOv8n integrates the Coordinate Attention mechanism into the C2f module to form the C2f-CA module, which strengthens the expression of inter-channel feature information, thereby improving the sensitivity to key areas and the ability to filter background interference. Secondly, the MLLAttention mechanism is introduced in the Neck part of the model to better deal with multi-scale problems. In addition, we have also innovatively improved the SPPF module to form the SPPF-GPE module, and enhanced the recognition effect of small and long-distance targets by combining global average pooling and global maximum pooling processing. The comprehensive application of these technologies is expected to enable the CAMLLA-YOLOv8n model to show higher accuracy and efficiency when detecting small-sized or long-distance human behaviors in surveillance videos.
Detecting dangerous human behavior does require specific adjustments to the model, such as optimizing data augmentation strategies and adjusting the configuration of the Attention mechanism. The effectiveness of these adjustments will depend on the diversity and quality of the training dataset, as well as whether the model can accurately capture and identify the key features of human behavior. Of course, the specific effects still need to be verified through actual data and scenarios.
4.How easy is to implement it in real time?
Response: Regarding the real-time implementation issue raised by the reviewer, we have tested the performance of the CAMLLA-YOLOv8n model on different hardware platforms to ensure that it can meet the requirements of real-time processing. Specifically, we first tested it on NVIDIA GeForce GTX 1080 Ti with the following hardware configuration: operating system Windows 10, CPU model AMD R7-5700X CPU 3.4GHz, GPU model NVIDIA GeForce GTX 1080 Ti, video memory 11GB, RAM 32GB DDR4 (16×2 GB), and SSD capacity 1TB. The test results show that the CAMLLA-YOLOv8n model can achieve real-time processing speed on 1080 Ti, meeting the requirements of high frame rate video surveillance. In addition, considering the application scenarios of edge computing devices, we also deployed and tested the model on the NVIDIA Jetson AGX Xavier edge development board and the HUAWEI Atlas 200l DK A2 development kit. On these edge devices, the CAMLLA-YOLOv8n model is also able to achieve real-time processing speed, which proves its good adaptability and reliability on edge devices of different performance levels.The original text is as follows:

Reviewer 2 Report
Comments and Suggestions for Authors
This study proposed an improved YOLOv8n Holstein cow behavior recognition method CAMLLA-YOLOv8n to improve the effect of cow behavior detection. The manuscript is somewhat innovative, the workload is adequate, and the experimental design is appropriate, but there is still some room for improvement.
Specific comments are as follows:
1. MLLAttention should be used in its full name on its first occurrence.
2. It is not necessary to retain five decimals for the model's evaluation metrics.
3. It is recommended that the optimal value for each metric in the table be bolded.
4. One presentation of the structural diagram of the CAMLLA-YOLOv8n will suffice.
5. Fig. 20 and Fig. 21 seem to be automatically generated image files during the model training process, and it is recommended to use the data not involved in the model training to do the visualization qualitative analysis of the model. It is suggested that the visualization figure should include the original image, GT, baseline and the improved model.
6. The manuscript presents too much science-based knowledge in the methods section and should highlight the innovation of the article.
7. The content of lines 78-80 is incoherent.
Author Response
Reviewer 2:
This study proposed an improved YOLOv8n Holstein cow behavior recognition method CAMLLA-YOLOv8n to improve the effect of cow behavior detection. The manuscript is somewhat innovative, the workload is adequate, and the experimental design is appropriate, but there is still some room for improvement.
Specific comments are as follows:
1.MLLAttention should be used in its full name on its first occurrence.
Response: Thanks to the reviewer for his suggestion. We have used the full name of MLLAttention "Mamba-Like Linear Attention" when it first appears in the article to ensure that all readers can understand it accurately. The original text is as follows:“(3) The Multi-layer Mamba-Like Linear Attention (MLLAttention) mechanism was introduced in the P3, P4, and P5 layers of the Neck of the YOLOv8n model to tackle the challenges posed by significant scale variations in cow behavior recognition.”
2.It is not necessary to retain five decimals for the model's evaluation metrics.
Response: We are very grateful for the reviewer's suggestions. Based on your feedback, we have adjusted the numerical precision of the evaluation indicators. To ensure the clarity of the experimental results while avoiding unnecessary complexity, we decided to adopt the principle of rounding off the last digit and retain the precision of all data indicators to four decimal places. This modification has been reflected in Tables 4 and 5 in the paper and the experimental results section of the paper. We believe that these adjustments will help readers understand the research data more intuitively while maintaining the scientificity and accuracy of the experimental results data.
3.It is recommended that the optimal value for each metric in the table be bolded.
Response: In order to more intuitively identify the optimal values of each performance indicator, we have bolded all the optimal values in Tables 4 and 5 of the experimental results section.
4.One presentation of the structural diagram of the CAMLLA-YOLOv8n will suffice.
Response: Thanks to the reviewer for his suggestion. The structure diagram of CAMLLA-YOLOv8n in this research paper is shown once. Specifically, Figure 6 shows the CAMLLA-YOLOv8n cow behavior recognition network architecture diagram in detail; Figure 7 shows the overall structure diagram of the unimproved YOLOv8n network for comparison; and Figure 10 specifically shows the Multi-level feature fusion and MLLAttention Mechanisms display of CAMLLA-YOLOv8n backbone network. These pictures are designed to clearly show the innovations we proposed and the comparison with the original model structure. We hope that this detailed display can help readers better understand the structure and improvements of the model.
5.Fig. 20 and Fig. 21 seem to be automatically generated image files during the model training process, and it is recommended to use the data not involved in the model training to do the visualization qualitative analysis of the model. It is suggested that the visualization figure should include the original image, GT, baseline and the improved model.
Response: Thanks to the reviewers for their suggestions. We have added Figure 20, and randomly selected 3 representative images from the test set for visual qualitative analysis to better evaluate the generalization ability of the model. In Fig.20, the first line is the original image, the second line is the manually annotated Ground Truth Box, the third line is the detection result of YOLOv8n, and the fourth line is the improved CAMLLA-YOLOv8n detection result. By comparison, CAMLLA-YOLOv8n shows significant advantages in handling occlusion between cows, dealing with multi-scale targets, and adapting to complex backgrounds, thus providing a clearer and more intuitive comparative analysis.The original text is as follows:
“In order to further verify the accuracy and detection effect of the CAMLLA-YOLOv8n model in the cow behavior recognition task, this study randomly selected three representative images from the test set for testing to demonstrate the model's performance in various scenarios. Fig.20 shows the test results. The first line is the original image, the second line is the manually annotated Ground Truth Box, the third line shows the detection results of YOLOv8n, and the fourth line is the improved CAMLLA-YOLOv8n detection results. By comparison, CAMLLA-YOLOv8n shows significant advantages in handling occlusions between cows, dealing with multi-scale targets, and adapting to complex backgrounds. Specifically, in the first scene, due to mutual occlusion between cows and the limitations of the monitoring range, the traditional YOLOv8n model struggles to identify partially occluded cows. For example, the standing cow partially exposed in the upper left is not detected, resulting in missed detection. Simultaneously, YOLOv8n’s detection accuracy for cows grazing on the lower left side is only 0.71, while CAMLLA-YOLOv8n’s detection accuracy reaches 0.91, demonstrating its superiority in handling scenes with occlusions and complex backgrounds. In the second scene, both models have high recognition accuracy for the cow standing in the middle, but YOLOv8n’s accuracy is 0.95, which is lower than CAMLLA-YOLOv8n’s 0.97. In the third scene, CAMLLA-YOLOv8n’s accuracy in identifying licking behavior reaches a perfect 1.00, slightly higher than YOLOv8n’s 0.99. For cows lying down in the distance, although the color of the ground and the fence may interfere with the detection, CAMLLA-YOLOv8n’s detection accuracy is still generally higher than YOLOv8n’s by about 0.02. In summary, the CAMLLA-YOLOv8n model proposed in this article shows better performance than the traditional model in terms of the diversity of cow behaviors, occlusion processing, and adaptability to background complexity, effectively improving the accuracy of cow behavior recognition.
In terms of real-time detection, we first tested the system on an NVIDIA GeForce GTX 1080 Ti with the following configuration: Windows 10 operating system, AMD R7-5700X CPU (3.4GHz), NVIDIA GeForce GTX 1080 Ti GPU, 11GB of video memory, 32GB DDR4 RAM (16×2 GB), and a 1TB solid-state drive. The average inference time of the CAMLLA-YOLOv8n model for processing a single image is 35.5ms, and the inference speed meets the requirements for real-time detection of video streams at 30fps. In addition, considering the application scenarios of edge computing devices, we also deployed and tested the model on the NVIDIA Jetson AGX Xavier edge development board and the HUAWEI Atlas 200 DK A2 development kit. On these edge devices, the CAMLLA-YOLOv8n model also achieves real-time processing speed, which demonstrates its good adaptability and reliability on edge devices of different performance levels.
During the training process of the CAMLLA-YOLOv8n model, the visualization results generated on dataset images, as shown in Fig.21 and Fig22, demonstrate that CAMLLA-YOLOv8n performs well in various challenging scenarios. It accurately identifies the behavior of cows in different environments, addressing the background complexity caused by Mosaic fusion and the increase in the number of cows, as well as the multi-scale, multi-behavioral characteristics and small targets. This effectively completes the recognition task of this study.”
6.The manuscript presents too much science-based knowledge in the methods section and should highlight the innovation of the article.
Response: We thank the reviewers for their valuable suggestions. We have made appropriate modifications to the methods section to reduce the introduction of basic scientific knowledge and focus on highlighting the innovation of the article and the unique advantages of our proposed method. In particular, we introduce the integrated application of Coordinate Attention Mechanism and Multi-layer Mamba-Like Linear Attention (MLLAttention) Mechanism in detail. The fusion of these two mechanisms not only enhances the model's ability to identify key features in complex scenes, but also significantly improves the accuracy of target detection. In addition, we also provide an in-depth explanation of the SPPF-GPE module. This module optimizes the model's detection capabilities for small targets by combining global average pooling and global maximum pooling, which is particularly important in practical applications. This modification can better demonstrate the contribution of this research and its innovative value in the field.
7.The content of lines 78-80 is incoherent.
Response: Thank you for the reviewer's suggestion. Regarding the incoherent content from lines 78 to 80, we have reviewed and reorganized this section to ensure the logical coherence and fluency of the expression. The original text is as follows:“Based on video image analysis technology, this method enables non-contact, automated, real-time online monitoring of cow behavior.”

Reviewer 3 Report
Comments and Suggestions for Authors
The research titled "CAMLLA-YOLOv8n: Cow Behavior Recognition Based on Improved YOLOv8n" aims to evaluate the effectiveness of an improved YOLOv8n-based method, CAMLLA-YOLOv8n, for recognizing Holstein cow behaviors in dairy farming. The subject of this article aligns well with the journal's focus and is, in my opinion, both highly interesting and timely. The application of technology in livestock farming is becoming an increasingly important topic, offering numerous opportunities for farmers to enhance animal welfare—a matter of growing public concern. Given the relevance of this issue, I suggest the authors incorporate a discussion of it in the paper's discussion section. Below are my comments for the authors:
General Comments:
- I recommend reformulating the title to make it clearer and more readable.
- I suggest the authors revise the simple summary to better align with the journal’s instructions for authors. It should clearly state the problem addressed, the aims and objectives, key results, conclusions, and how these findings are valuable to society. Furthermore, it should be written in a way that is accessible to a general audience.
- I advise the authors to rewrite the abstract to improve readability.
- Avoid using keywords that are already present in the title.
- Although the introduction is well-structured, it is quite lengthy. I suggest the authors condense it by moving parts of the content to the discussion section.
- In the materials and methods section, I recommend including information on the characteristics of the study subjects and farms (even as supplementary material) to enhance the study's reproducibility. Additionally, consider including some details about farm management practices.
- It would also be interesting to add a descriptive statistics section on the farms in the results section to better contextualize the research.
- Enrich the discussion by including the study's limitations and its practical applications.
Specific Comments:
LL 40-41: Please include a citation. For example, I suggest using DOI: 10.3390/ani14162367.
L 53: I recommend the authors review the journal’s guidelines carefully and correctly format the citations.
Citation n.3 is somewhat outdated; I suggest focusing on more recent papers, such as DOI: 10.1080/1828051X.2022.2032850.
L 61: I suggest replacing "dense" with "large scale."
LL 66-68: I disagree with this statement. Technological advances have significantly reduced the impact on animals (a different case would be extensively raised animals). If the authors want to highlight the benefits of image analysis for monitoring, I suggest discussing how using collars or other sensors can be problematic for farmers. Monitoring their functionality can be challenging, they may not provide accurate data if improperly positioned, and they are prone to breakage, especially in younger animals that move more.
LL 70-71: I also disagree with this statement. Many companies now offer monitoring technologies that measure various parameters with a single device.
Author Response
Reviewer 3:
The research titled "CAMLLA-YOLOv8n: Cow Behavior Recognition Based on Improved YOLOv8n" aims to evaluate the effectiveness of an improved YOLOv8n-based method, CAMLLA-YOLOv8n, for recognizing Holstein cow behaviors in dairy farming. The subject of this article aligns well with the journal's focus and is, in my opinion, both highly interesting and timely. The application of technology in livestock farming is becoming an increasingly important topic, offering numerous opportunities for farmers to enhance animal welfare—a matter of growing public concern. Given the relevance of this issue, I suggest the authors incorporate a discussion of it in the paper's discussion section. Below are my comments for the authors:
General Comments:
- I recommend reformulating the title to make it clearer and more readable.
- I suggest the authors revise the simple summary to better align with the journal’s instructions for authors. It should clearly state the problem addressed, the aims and objectives, key results, conclusions, and how these findings are valuable to society. Furthermore, it should be written in a way that is accessible to a general audience.
- I advise the authors to rewrite the abstract to improve readability.
- Avoid using keywords that are already present in the title.
- Although the introduction is well-structured, it is quite lengthy. I suggest the authors condense it by moving parts of the content to the discussion section.
- In the materials and methods section, I recommend including information on the characteristics of the study subjects and farms (even as supplementary material) to enhance the study's reproducibility. Additionally, consider including some details about farm management practices.
- It would also be interesting to add a descriptive statistics section on the farms in the results section to better contextualize the research.
- Enrich the discussion by including the study's limitations and its practical applications.
Response: We are very grateful for the reviewers' suggestions, which are very helpful in improving the quality of our paper. We rewrote the simple abstract according to the reviewers' suggestions. In the revised simple abstract, we clarified the research question, research aims and objectives, and results, and emphasized the social value of these findings. We rewrote the abstract comprehensively according to the suggestions, enhanced its logic and flow, and ensured that the key information was clearly communicated. We tried to use easy-to-understand language so that non-specialist readers can understand it.
Specific Comments:
1.LL 40-41: Please include a citation. For example, I suggest using DOI: 10.3390/ani14162367.
Response: We are very grateful for the reviewer's suggestion. We have reviewed and cited the literature you recommended (DOI: 10.3390/ani14162367.) to support our argument that studying animal behavior is crucial to understanding how animals interpret and respond to the environment. The literature provides strong support and further strengthens our argument in this paragraph. The original text is as follows:“In poultry farming, studying animal behavior is crucial for understanding how animals perceive and respond to their environment.”
2.L 53: I recommend the authors review the journal’s guidelines carefully and correctly format the citations.
Response: We thank the reviewer for his suggestions. We carefully read and followed the journal's guidelines, checked the citation format in the paper one by one, and corrected at least three citation formats that did not meet the journal's standards, namely citations n.3, n.4, n.5, n.6, n.8 and n.9, to ensure that they met the journal's citation standards.
3.Citation n.3 is somewhat outdated; I suggest focusing on more recent papers, such as DOI: 10.1080/1828051X.2022.2032850.
Response: We thank the reviewer for his valuable suggestions. We have considered the reference you recommended (DOI: 10.1080/1828051X.2022.2032850), but after careful review, we believe that retaining the current reference can better support the theme and argument of our article, so we decided not to change the reference. We always attach great importance to the selection and use of literature to ensure that it can best support the argument of our research.
4.L 61: I suggest replacing "dense" with "large scale."
Response: We thank the reviewer for the suggestion. We have replaced the word “dense” with “large scale” to express our meaning more accurately.
5.LL 66-68: I disagree with this statement. Technological advances have significantly reduced the impact on animals (a different case would be extensively raised animals). If the authors want to highlight the benefits of image analysis for monitoring, I suggest discussing how using collars or other sensors can be problematic for farmers. Monitoring their functionality can be challenging, they may not provide accurate data if improperly positioned, and they are prone to breakage, especially in younger animals that move more.
Response: Thank you for the reviewer's suggestion. We have reviewed this paragraph, considering that current technological advances have greatly reduced the impact on animals, and focused on discussing the challenges that collars or sensors may face in practical applications, especially the problems that may arise when placing and using young animals, to ensure the comprehensiveness and objectivity of the viewpoint. The original text is as follows:“Using contact-based sensors to monitor cow behavior data also has its drawbacks. For instance, these sensors typically require animals to wear them in specific places such as collars or ankles to collect movement and physiological data for behavior identification. However, employing contact-based devices poses challenges, improper placement may result in inaccurate sensor data. Moreover, these devices are prone to damage, especially in younger animals that are more active, impacting the reliability and continuity of monitoring.”
6.LL 70-71: I also disagree with this statement. Many companies now offer monitoring technologies that measure various parameters with a single device.
Response: Thank you for your valuable comments. The question you raised is very important, so we have removed the description of the single function of the monitoring equipment, that is, "In actual farming, if you want to monitor multiple behaviors, you need to wear multiple sensors with different functions." After this adjustment, the discussion is more accurate and clear. The original text is as follows:“Using contact-based sensors to monitor cow behavior data also has its drawbacks. For instance, these sensors typically require animals to wear them in specific places such as collars or ankles to collect movement and physiological data for behavior identification. However, employing contact-based devices poses challenges, improper placement may result in inaccurate sensor data. Moreover, these devices are prone to damage, especially in younger animals that are more active, impacting the reliability and continuity of monitoring.”

Reviewer 4 Report
Comments and Suggestions for Authors
It is a practical study conducted on a cow farm, utilizing YOLOv8 to analyze Holstein cow behavior. However, there are some suggestions before publication:
1. In lines 132-166, please summarize your objectives more concisely.
2. In Figure 1, the camera image is unclear.
3. Please provide sample pictures of each behavior in Table 2.
4. In Figure 6, I could not find the output picture, similar to Figure 7.
5. The font size in Figure 18 is too small.
Author Response
Reviewer 4:
It is a practical study conducted on a cow farm, utilizing YOLOv8 to analyze Holstein cow behavior. However, there are some suggestions before publication:
1.In lines 132-166, please summarize your objectives more concisely.
Response: Thank you for the reviewer's suggestion. To improve the clarity of the discussion, we have reorganized and simplified the content from lines 132 to 166, which mainly introduces the contribution of this paper. Now this paragraph is more direct and concise, so that readers can understand our research results more clearly. The original text is as follows:“(1) We installed high-definition cameras at the sixth dairy farm of Tianjin Jialihe Animal Husbandry Group Co., Ltd. in Baodi District, Tianjin, China, and collected video data of Holstein cows' daily behavior for about 110 days. After removing redundant frames, we collected a total of 2418 images, which were annotated into 7 behavior categories based on expert classification, with 23073 boxes labeled using the CVAT tool. (2) To propose an improved YOLOv8n-based behavior recognition method for Holstein cows, named CAMLLA-YOLOv8n, which integrates the Coordinate Attention mechanism into the C2f module to form the C2f-CA module. This enhances the model's ability to accurately recognize spatial relationships between different cow positions, focus on key areas, and filter background interference. (3) The Multi-layer Mamba-Like Linear Attention (MLLAttention) mechanism was introduced in the P3, P4, and P5 layers of the Neck of the YOLOv8n model to tackle the challenges posed by significant scale variations in cow behavior recognition. (4) The SPPF-GPE module was formed by improving the SPPF module, combining global average pooling and global maximum pooling to enhance the model's ability to cope with environmental changes and capture key features of cow behavior. (5) Considering the limitations of traditional IoU loss in cow detection, we introduced Shape-IoU to focus on the shape and scale features of the bounding box, improve the alignment between prediction and ground truth boxes, and enhance detection accuracy.”
2.In Figure 1, the camera image is unclear.
Response: Thank you for the reviewer’s suggestion. We have updated and replaced the images in Figure 1, ensuring that they are clear enough to better convey the necessary information.
3.Please provide sample pictures of each behavior in Table 2.
Response: Thanks to the reviewer for his suggestion. We have shown examples of cow behavior sample images included in the dataset in Figure 2 so that readers can better understand the characteristics of each cow behavior listed in Table 2. The original text is as follows:“Six behaviors of dairy cows, including grazing, standing, lying, licking, mating and fighting, and the empty state of the cow bed were selected as the research objects. These behaviors are closely related to the health evaluation of dairy cows. Sample pictures of cow behavior in the dataset are shown in Fig.2.”
4.In Figure 6, I could not find the output picture, similar to Figure 7.
Response: Thanks to the reviewer for his suggestion. The "Detect" output part in Figure 6 is located at the end of the network. It is the model output area, which outputs the final bounding boxes, classe, and confidence scores of the input image. In order to enhance the visibility and recognition of this area, we enlarge the font and adjust its color to make it more prominent in the figure.
5.The font size in Figure 18 is too small.
Response: Thank you for the reviewer’s suggestion. We have adjusted the font size in Figure 18 to ensure that the data and information in the figure are clearer and easier to read.

Round 2
Reviewer 2 Report
Comments and Suggestions for Authors
All my concerns have been resolved in the revision.Just a minor comment.
1. Some of the values in Figure 18 are occluded from each other.
Author Response
Thank you for the reviewer's suggestion. According to the reviewers' valuable suggestions, I have adjusted the positioning and sizing of the labels in Figure 18 to ensure that the values are no longer occluded. The revised figure now clearly displays all the data without any overlap, improving its readability and effectiveness in conveying the intended information. Thank you for your guidance in enhancing the presentation of my paper. The updated image is shown below:

Reviewer 3 Report
Comments and Suggestions for Authors
I am happy with the changes introduced after my request of clarification. I have just a few minor comments to offer:
L 43: I suggest replacing "poultry farming" with "livestock farming."
LL 60-62: I would rephrase as, “Cow behavior detection methods can be categorized into three main types: traditional staff observation, contact-based sensor detection, and non-contact image recognition.”
L 80: I recommend removing the term "online."
Author Response
1.L 43: I suggest replacing "poultry farming" with "livestock farming."
Response: Thank you for the reviewer's suggestion. I have changed "poultry farming" to "livestock farming" to more accurately reflect the content of the research.
2.LL 60-62: I would rephrase as, “Cow behavior detection methods can be categorized into three main types: traditional staff observation, contact-based sensor detection, and non-contact image recognition.”
Response: Thank you for the reviewer's suggestion. I have rephrased according to your guidance as: "Cow behavior detection methods can be categorized into three main types: traditional staff observation, contact-based sensor detection, and non-contact image recognition." This expression is clearer and more precise.
3.L 80: I recommend removing the term "online."
Response: Thank you for the reviewer's suggestion. I have removed the term "online" to make the description more concise and clear.
